# Seasonal features and origins of carbonaceous aerosols at Syowa Station, coastal Antarctica

Keiichiro Hara[1], Kengo Sudo[2], Takato Ohnishi[2], Kazuo Osada[2], Masanori Yabuki[3], Masataka Shiobara[4], Takashi Yamanouchi[4]

[1] Department of Earth System Science, Faculty of Science, Fukuoka University, Fukuoka, 814-0180, Japan
[2] Graduate School of Environmental Studies, Nagoya University, Nagoya, 464-8601, Japan
[3] Research Institute for Sustainable Humanosphere, Kyoto University, Kyoto, 611-0011, Japan
[4] National Institute of Polar Research, Tokyo, 190-0014, Japan

*Correspondence to:* Keiichiro Hara (harakei@fukuoka-u.ac.jp)

**Abstract.** We have measured black carbon (BC) concentrations at Syowa Station, Antarctica since February 2005. The measured BC concentrations in 2005–2016 were corrected to equivalent BC (EBC) concentrations using Weingartner's method. Seasonal features of EBC concentrations, long-range transport from mid-latitudes to the Antarctic coast, and their origins were characterized. Results show that daily median EBC concentrations were below the detection limit (0.2 ng m$^{-3}$) to 63.8 ng m$^{-3}$ at Syowa Station (median, 1.8 ng m$^{-3}$; mean, 2.7 ng m$^{-3}$ during the measurement period of February 2005 – December 2016). Although seasonal features and year-to-year variations of EBC concentrations were observed, no long-term trend of EBC concentrations was clear during our measurement period. Seasonal features of EBC concentrations showed a spring maximum during September–October at Syowa Station. To elucidate EBC transport processes, origins, and the potential source area (PSA), we compared EBC data to backward trajectory analysis and chemical transport model simulation. From comparison with backward trajectory, high EBC concentrations were found in air masses from the marine boundary layer. This finding implies that transport via the marine boundary layer was the most important transport pathway to EBC concentrations at Antarctic coasts. Some EBC was supplied to the Antarctic region by transport via the upper free troposphere. Chemical transport model simulation demonstrated that the most important origins and PSA of EBC at Syowa Station were biomass burning in South America and southern Africa. Fossil fuel combustion in South America and southern Africa also have important contributions. The absorption Ångström exponent (AAE) showed clear seasonal features with 0.5–1.0 during April–October and maximum (1.0–1.5) in December–February. The AAE features might be associated with organic aerosols and mixing states of EBC.

## 1 Introduction

Carbonaceous aerosols are major aerosols in the troposphere (e.g., Gelencsér, 2004; Gilardoni, and Fuzzi, 2017). In general, carbonaceous aerosols include organic compounds and particulate graphite (e.g., Gelencsér, 2004; Andreae and Gelencsér, 2006; Bond et al., 2013). Various terms such as elemental carbon (EC), black carbon (BC), organics, and soot are used to describe carbonaceous particles. Apart from secondary organics associated with biogenic cycles, most of carbonaceous aerosols (e.g. soot) can be released from combustion of biomass and fuels. Soot particles consist of refractory and insoluble matter (aka EC) and organics (e.g., Andreae and Gelencsér, 2006). As defined by Novakov (1984), BC comprises particulate graphitic particles. Recently, BC has been defined by the following physical properties: (1) strong light absorption, (2) refractory, (3) insoluble, and (4) including aggregates of small carbon spherules (e.g., Bond et al., 2013). Because of its strong optical absorption, BC has been a concern for atmospheric radiation budgets and climate effects (e.g., Bond et al., 2013, references therein). In addition to BC, mineral particles containing iron oxides (e.g., hematite and magnetite) and some organic aerosols (e.g., brown carbon, BrC) have light absorption in visible and ultraviolet (UV) spectral bands (Bond et al., 2013, references therein; Moteki et al., 2017). Furthermore, BC can alter surface albedo after deposition onto snow surfaces in polar regions (e.g., Flanner et al., 2007; Aoki et al., 2011; Hadley and Kirchstetter, 2012; Bond et al., 2013). In the Antarctic region,

BC effects on radiation budgets are regarded as negligible because of low BC concentration (e.g., Bodhaine, 1995; Weller et al., 2013).

Atmospheric BC is released directly from incomplete combustion processes. The Antarctic region is isolated from large combustion sources related to human activities at low latitudes and mid-latitudes. Therefore, local origins of BC in the Antarctic area are limited to (1) human activity at research stations, (2) usage of snow vehicles for travel, (3) operations of airplanes and research vessels during summer, and (4) ship-borne tourism mainly on the Antarctic Peninsula (Shirasat and Graf, 2009; Graf et al., 2010). Although local contamination from these sources can engender temporarily high BC concentrations (e.g., Wolff and Cachier, 1998; Hansen et al., 2001; Hagler et al., 2008), the BC source strength is likely to be negligible or only slight throughout the Antarctic region. Indeed, earlier work has shown that BC concentrations are lower at higher latitudes (Wolff and Cachier, 1998; Weller et al., 2013). It has been considered that BC must be supplied from outside of Antarctica, i.e. long-range transport, to maintain the background BC level and that it has seasonal features in the Antarctic atmosphere because of the low BC source strength in the Antarctic region. In other words, BC in the Antarctic atmosphere is useful as a tracer of atmospheric substances derived from combustion processes occurring at mid-latitudes and low latitudes.

Earlier studies (Wolff and Cachier, 1998; Weller et al., 2013) have pointed out the likelihood that BC in the Antarctic atmosphere originates from biomass burning. Additionally, BC is transported directly from South America (Fiebig et al., 2009; Hara et al., 2010) and from southern Africa (Hara et al., 2010) by poleward flow associated with cyclone activity in the Southern Ocean. Additionally, Weller et al. (2013) and Perreira et al. (2006) demonstrated that BC at Neumayer and Ferraz is supplied from biomass burning in South America. Considering BC outflow from South America and southern Africa and local emission from tourism, we must consider longitudinal distribution of BC in Western and Eastern Antarctica to compare and elucidate BC concentrations measured at each station and transport pathway to the Antarctica. Levoglucosan, as a tracer of biomass burning, was detected in aerosols and snow taken in the Antarctica (Gambaro et al., 2008, Hu et al., 2013; Zangrando et al., 2016). Therefore, the BC must be transported from areas where the biomass burning occurred. In fact, BC records for the past 150 years in the Antarctic ice cores (WAIS core in Western Antarctica and Low dome core in Eastern Antarctica) showed influences by El Niño-Southern Oscillation (ENSO) and BC emissions from biomass burning and human activity in the source areas (Bisiaux et al., 2012). Furthermore, high correlation between BC and $NH_4^+$ in the Antarctic ice core indicates BC and $NH_4^+$ from primary sources of biomass burning (Pasteris et al., 2014). Certainly, BC measurements taken during a few decades at Neumayer indicate an unclear long-term trend of BC concentrations, although a slight decreasing trend is apparent in summer (Weller et al., 2013).

In contrast to the Arctic atmosphere, earlier works concluded that anthropogenic effects were only slight and negligible for aerosols in the Antarctic atmosphere (e.g., Weller et al., 2011, 2013), although some anthropogenic metals such as Pb have been found in snow and ice cores in the Antarctic region (Planchon et al., 2002; Vallelonga et al., 2002). Considering that biomass burning occurs on the ground in forests and grasslands, anthropogenic BC (derived mainly from fossil fuel combustion) can be transported to Antarctica. With the recent intense economic development of countries of the Southern Hemisphere, the contributions of anthropogenic BC must be assessed. Nevertheless, the contributions of biomass burning and anthropogenic processes to BC concentrations in the Antarctic troposphere and their potential source area (PSA) have been neither quantitatively analyzed nor discussed in the relevant literature.

To elucidate BC transport from the low latitudes and mid-latitudes to the Antarctic region, we must ascertain the potential source area (PSA) and transport pathway. Actually, BC cannot be vaporized in ambient conditions. Therefore, BC must be transported from the origins (i.e. combustion processes) to the Antarctica. However, chemical analyses such as isotope ratio

investigations are difficult to apply for identification of BC origins because the major BC component is graphite. Hara et al. (2010) described BC transport from South America and southern Africa to Syowa Station, Antarctica. Similarly to BC, mineral particles are transported from their origins to Antarctica, except for local emissions originating within the Antarctic Circle. For identification of the origins of mineral particles, earlier studies have been conducted to analyze and assess PSA of mineral particles based on Nd/Sr isotope ratios (Smith et al., 2003; Delmonte et al., 2004, 2008; Bory et al., 2010; Valleloga et al., 2010; Aarons et al., 2016), Pb isotope ratios (De Deckker et al., 2010; Gilli et al., 2016), rare earth element patterns (Gabrielli et al., 2010; Valleloga et al., 2010; Wegner et al., 2012, Aarons et al., 2016), and trajectory/models (Perreira et al., 2004; Li et al., 2008; Albani et al., 2010; Gasso et al., 2010; Krinner et al., 2010; Neff and Bertler, 2015). From the aspect of mineral particles transported into Antarctica, South America (mostly Patagonia) has been identified as the most dominant PSA, whereas Australia and Africa respectively show minor and unimportant PSAs (e.g., Neff and Bertler, 2015). Although one must consider the following differences between BC and minerals, (1) geographical locations of PSA, (2) seasonality of source strength, and (3) size of aerosol particles containing BC and minerals, BC can be transported by outflow from the continents in the mid-latitudes to Antarctica. Here, we combine BC measurements with backward trajectory and chemical transport model simulation. This study was conducted to elucidate BC origins and PSA and to characterize BC concentrations and their seasonal features at Syowa Station, Antarctica located in the Indian Ocean sector.

## 2. Measurements, modelling and analysis

### 2.1 Aerosol measurements at Syowa Station, Antarctica

Aerosol measurements were conducted as part of the Japanese Antarctic Research Expedition (JARE) at Syowa Station on eastern Ongul Island, Antarctica (69°00′S, 39°35′E, ca. 29 m above sea level), located as presented in Fig. 1. To Syowa, the icebreaker ship Shirase approaches every summer (mainly end-December – early February) for the transportation of fuel and materials to support wintering operations and scientific activity. Some airplanes and helicopters operate occasionally during summer and not during other seasons. In contrast to the situation on the Antarctic Peninsula, ship-borne tourism was not done off Syowa during our measurements because of the station's distance from the other continents and the long distance between Syowa and the sea-ice margin (ca. 100 km even in summer). The BC concentrations have been measured using a multi-wavelength aethalometer (AE31; Magee Scientific) since February 2005, although the aethalometer measurements were not taken from January 2007 through January 2008 because of instrumental troubles. For this study, we used BC data measured in 2005–2016. The wavelengths of light sources in the aethalometer were 370, 470, 520, 590, 660, 880, and 950 nm. The aethalometer was operated in a clean air observatory located on the windward side of prevailing winds, ca. 400 m distant from the main area of Syowa, where a diesel power station was operating. The aethalometer operated under the following conditions: flow rate (ca. 9 L min$^{-1}$ in 2005–2006 and ca. 11 L min$^{-1}$ in 2008–present), data record resolution of 15 min, and spot change every 24 hr. For this study, we used a high-sensitivity AE31 instrument. The area of a circular spot to accumulate aerosols on the filter tape was 60.3 $\pm$ 1 mm$^2$.

In an aethalometer, BC concentrations are measured by light attenuation resulting from optical absorption of BC collected on the filter tape. As earlier works have suggested (e.g., Weingartner et al., 2003; Bond et al., 2013), the filter-based BC measurements have scattering and shadowing effects that can engender error of BC measurements. Therefore, we used Weingartner's procedures for this study to correct BC concentrations (Weingartner et al., 2003). Details of procedures for correction of BC concentrations are presented in *Supplementary Information.* Light attenuation and optical attenuation coefficients in UV and visible ranges can be influenced greatly by optical absorption of organics and mineral dusts (e.g., Bond et al., 2013). Attenuation at 880 nm is used widely for BC retrievals. Therefore, for this study, we used attenuation data in IR channel ($\lambda$ = 880 nm) to estimate the BC concentrations. As reported by Bond et al. (2013), hereinafter we use the term of equivalent BC (EBC) for the corrected BC mass concentrations and the measured BC concentrations using filter-base optical

techniques from earlier works. The detection limit of EBC in the aethalometer depends on the optical signal-to-noise ratio. We checked the optical signal-to-noise ratios of aerosol-free conditions several times in our measurement conditions. The detection limit was estimated as 0.2 ng m$^{-3}$ in the IR channel under our measurement conditions ($\Delta t$ = 120 min and flow rate, 10 L min$^{-1}$). Uncertainty of the measured EBC concentrations relates to (1) stability of the optical signal, (2) flow rate control, (3) spot

area, and (4) scattering and shadowing effects. The detection limit value corresponds to uncertainty resulting from processes of (1)–(3). Uncertainty by the process (4) depends on the aerosol number concentrations and optical properties (single scattering albedo). The EBC concentrations corrected using Weingartner's method were mostly lower by 0.5–2% compared to the uncorrected EBC concentrations in this study (Fig. S1). Less difference between the corrected and uncorrected EBC concentrations might derive from higher single scattering albedo and replacement of the filter spot before optical attenuation

reaching to 10% in most cases in our measurement conditions at Syowa.

Using multi-wavelength optical absorption values by aerosols retrieved from aethalometer, we estimated the absorption Ångström exponent (AAE) in this study. The values of AAE can be represented as

$C_{abs}(\lambda) = b_{abs}\lambda^{-AAE}$,                                                                    (1),

where $C_{abs}$, and $b_{abs}$ respectively represent the optical absorption cross section at the $\lambda$ (wavelength) and optical absorption coefficient. First, $b_{abs}$ or $b_{ATN}$ (optical attenuation coefficient) must be known to estimate AAE (Supplementary). Then, we estimated AAE in UV–IR range (370–950 nm) and the visible (Vis)–IR range (590–950 nm) to elucidate the effects of organics

and EBC onto optical absorption properties.

When winds come from the main area of the station, aerosol data can be contaminated considerably. Before data analysis and discussion, BC data were screened using wind data (direction and speed) provided from the Japan Meteorological Agency and using condensation nuclei (CN) concentrations obtained from JARE aerosol monitoring data by the following procedures.

First, we estimated 10-min-mean CN concentrations from raw CN data with 1 min resolution to identify local contamination events. When wind flowed from contaminable sector (wind direction of 180–330°) and the wind speed was less than 2 m s$^{-1}$, the 10-min-mean CN data were removed as "locally contaminated data". Furthermore, the 10-min-mean CN data were screened also in cases of relative standard deviation larger than 10% in 10-min-mean estimation under conditions with wind speed less than 15 m s$^{-1}$ because of  the likelihood of local contamination from moving contamination sources such as snow

vehicles. For wind speeds greater than 15 m s$^{-1}$ corresponding to storm conditions, the CN data  were retained as "non-locally contaminated". Operation of snow vehicles was not permitted during strong winds because of safety guideline at Syowa. Additionally, stronger winds came from the prevailing wind direction (mainly 0–80°: clean air sector). Therefore, local contamination was not included in cases of stronger winds. Then, EBC and AAE data were filtered using the screened CN data. When local contamination was identified in CN data within 2 hr, the EBC data were removed as "locally contaminated".


## 2.2 Analysis of air mass history and origins

For this study, the 120-hr backward trajectory was computed to elucidate the transport pathway and origins of air masses transported to Syowa. The backward trajectory was calculated using the model vertical velocity mode in the NOAA-HYSPLIT model with meteorological data of NCEP reanalysis (Stein et al., 2015). The initial point was at an altitude of 500 m above

ground level over Syowa, Antarctica. For comparison between hourly mean EBC concentrations and the air mass history, the backward trajectory was calculated every hour from January 2005 through December 2016 in this study. Here, we use the following criteria to divide each air mass origin: marine, <66°S; coastal, 66–75°S; Antarctic-continental, >75°S; boundary

layer (BL), <1500 m; free troposphere (FT), >1500 m. Then, times passing in each area such as marine BL (MBL), coastal BL, continental BL, continental FT, coastal FT, and marine FT (MFT) were counted for each backward trajectory. The areas with air masses staying for the longest times in the 5-day backward trajectory were identified as air mass origins.

## 2.3 CHASER (MIROC-ESM) Model

Chemical atmospheric global climate model for studies of atmospheric environment and radiative forcing (CHASER) (Sudo et al., 2002; Sudo and Akimoto, 2007), developed mainly at Nagoya University and the Japan Agency for Marine-Earth Science and Technology (JAMSTEC), is a coupled chemistry climate model (CCM) simulating atmospheric chemistry and aerosols. Aerosols are examined using the Spectral Radiation-Transport Model for Aerosol Species (SPRINTARS) module (Takemura et al., 2005). It has been developed also in the framework of the Model for Interdisciplinary Research On Climate

(MIROC) – Earth System Model (ESM), MIROC-ESM-CHEM (Watanabe et al., 2011). CHASER simulates details of chemistry in the troposphere and stratosphere with an online aerosol simulation including the production of particulate nitrate and secondary organic aerosols. As a standard configuration, the model's horizontal resolution is selected as T42 ($2.8° \times 2.8°$), with 57 layers extending vertically from the surface up to about 55 km altitude. Regarding the overall model structure, CHASER is fully coupled with the climate model core MIROC, permitting atmospheric constituents (both gases and aerosols)

to interact radiatively and hydrologically with meteorological fields in the model. The chemistry component of CHASER includes consideration of the $O_x$-$NO_x$-$HO_x$-$CH_4$-CO chemical system with oxidation of non-methane volatile organic carbons (NMVOCs), halogen chemistry, and the $NH_x$-$SO_x$-$NO_3$ system. In all, 96 chemical species and 287 chemical reactions are considered. In the model, primary NMVOCs include $C_2H_6$, $C_2H_4$, $C_3H_8$, $C_3H_6$, $C_4H_{10}$, acetone, methanol, and biogenic NMVOCs (isoprene, terpenes). For the present study, CHASER uses interannually constant anthropogenic emissions

(EDGAR-HTAP2-2008, http://edgar.jrc.ec.europa.eu/htap_v2/) with the biomass burning emission dataset (MACC reanalysis). Regional biomass burning emission and its seasonal trend in MACC are similar to those in other inventory (e.g. GFED), although slight difference exist in regional distribution of biomass burning in each inventory. The model was nudged to the NCEP FNL ds083 (u, v, T) and HadiSST/ICE (2000–2017).

The aerosol component of CHASER considers BC tracers of two types: hydrophobic BC (in external mixture) and hydrophilic BC (internally mixed with water-soluble species such as organics or $SO_4^{2-}$). In the latest model version, the aging process of BC in which hydrophobic BC is converted gradually to hydrophilic is simulated considering the condensation of sulfuric acid ($H_2SO_4$) and semi-volatile organic carbons onto the BC surface, and coagulation of BC with water-soluble particles (organics, $SO_4^{2-}$, etc.). The current model configuration calculates BC aging with time constants of less than one day in the PBLs, and a

few days or weeks in the free troposphere depending on the abundances in $SO_2$, volatile organic carbons, and water-soluble aerosols.

For this study, a tagged BC tracer simulation is newly introduced into CHASER for estimating the respective contributions from different regions and types of emission to the long-range transport of BC. The tagged BC simulation, performed basically

in the same framework of the tagged $O_3$ simulation developed by Sudo and Akimoto (2007), separates the globe into 15 regions as presented in Fig. 2 and calculates transport and deposition of BC emitted from the regions as distinct tracers. For the individual BC tracers, we also discriminate different emission sectors: (1) biomass burning, (2) fossil-fuel combustion, and (3) others (such as cooking and open burning). The tagged BC emissions are first injected into the atmosphere as hydrophobic BC (in external mixture). They then undergo aging processes to be converted to hydrophilic BC as described above. To avoid

confusion of the term (e.g. EBC), we use the term of mBC hereinafter to designate the simulated mass BC concentrations.

## 3. Results and Discussion

### 3.1 Variations of EBC at Syowa Station, Antarctica

Figure 3 depicts seasonal features of EBC concentrations at Syowa Station, Antarctica from February 2005 through December 2016. In this study, median EBC concentrations are used for discussion because the mean EBC concentrations can be overestimated relative to ambient EBC concentrations without local contamination when the unfilterable data derived from local contamination were present in our data screening procedures. Daily median EBC concentrations ranged from below the detection limit (<0.2 ng m$^{-3}$) to 63.8 ng m$^{-3}$ during the measurement period. Modal, median, and mean concentrations were, respectively, 1.1, 1.8, and 2.7 ng m$^{-3}$ (Fig. S2 in Supplementary Materials). In addition, the distributions of EBC concentrations were approximated by lognormal distributions ($R^2$ = 0.9983) as

$$\text{F}[EBC] = ae^{0.5\left(\frac{ln(EBC - EBC_0)}{b}\right)^2}, \tag{6}$$

where $a$, $b$, and $EBC_0$ respectively stand for 529.75, 0.7270, and 1.12. High EBC concentrations were often observed in winter–spring during 2005–2009. Measurement conditions (e.g., tube length and room temperature) and analytical procedures were the same from 2005–2016. Therefore, this change might result from variations of frequency or strength of EBC transport events rather than measurement and analytical reasons.

From trend analysis (Supplementary and Fig. S3), a very slight decreasing trend (-0.036 ng m$^{-3}$ yr$^{-1}$, p = 0.0145) was observed in our measurements for 2005–2016. However, an increasing trend (0.105 ng m$^{-3}$ yr$^{-1}$, $P$ < 0.001) was obtained in 2010–2016. These trend values included temporal trends, as explained below. Therefore, we concluded only slightly whether these trends were long-term EBC trends or not. More continuous EBC measurements must be taken at Syowa Station to analyze long-term trends. Although a decreasing trend of EBC concentrations in summer (November and December) was found at Neumayer (Weller et al., 2013), no seasonal long-term trend was clear at Syowa except for July (Fig. S4 in *Supplementary Information*). At a glance, EBC concentrations in July showed an increasing trend for 2011–2016 (0.325 ng m$^{-3}$ yr$^{-1}$ in monthly median and 0.363 ng m$^{-3}$ yr$^{-1}$ in monthly mean). However, we must consider the likelihood that EBC concentrations in winter (June–August) declined in 2010–2012 rather than following the increasing trend by EBC emissions at middle and low latitudes. Indeed, this variation in July might be related to changes of air mass origins (details are discussed in section 3.2).

Before comparison between our EBC data and EBC concentrations measured at the other Antarctic stations, we must consider observation procedures and data quality of EBC in earlier works. Although different instruments were used for EBC measurements among Syowa (7 wavelength aethalometer, AE31), Halley (aethalometer, AE10), Neumayer (aethalometer: AE10 and Multi-angle absorption photometer (MAAP)), and South Pole (particulate soot absorption photometer (PSAP)), the EBC measurement principles were similar (i.e. filter-base optical attenuation measurement). In earlier works, EBC concentrations were uncorrected, unlike this study. The EBC concentrations corrected using Weingartner's method decreased mostly by 0.5–2% compared to the uncorrected EBC concentrations in this study (Fig. S2). The lesser difference between the corrected and uncorrected EBC concentrations might result from (1) higher single scattering albedo and (2) replacement of filter spot before optical attenuation reaching to 10% in most cases in our measurement conditions at Syowa. Therefore, we can compare EBC concentrations in this study to the uncorrected EBC concentrations measured at other Antarctic stations in previous works. In addition to filter-based EBC measurements, a single particle soot photometer (SP2) has been used for the measurement of refractory BC (rBC) (e.g., Bond et al., 2013; Sharma et al., 2017). According to Sharma et al. (2017), high correlation with $R^2$ = 0.8–0.9 and slopes = 1.2–1.6 was observed between rBC and EBC in aerosols in the Arctic, where aerosol concentrations and anthropogenic effects were greater and stronger than those in Antarctica. Considering different conditions

of aerosol chemistry and optical properties between those in Antarctica and the Arctic, correlation in the Antarctica is expected to be different from that in the Arctic. No report of the relevant literature has described SP2 used to measure rBC year-round in the Antarctic region. Because of higher single-scattering albedo and lower aerosol concentrations in the Antarctica, differences between rBC and EBC might not be greater than in the Arctic.

The EBC concentrations were similar to the EBC concentrations measured at coastal stations such as Halley and Neumayer (Wolff and Cahier, 1998; Weller et al., 2013). In contrast, EBC concentrations at Ferraz, Maitri, and Larsemann Hills were higher than those at Syowa (Pereira et al., 2006; Chaubey et al., 2010). Ferraz is located in the northern area of the Antarctic Peninsula as presented in Fig. 1. Air masses at Ferraz were transported frequently from South America (Pereira et al., 2004),

so that the long-range transport from South America might engender higher EBC concentrations at Ferraz than those at other Antarctic coast locations. The EBC concentrations at Maitri and Larsemann Hills during summer were markedly higher than those at Syowa and Neumayer. Considering the geographical locations of these stations, as presented in Fig. 1, the high EBC concentrations at Maitri and Larsemann Hills might result from insufficient screening of data contaminated locally by human activity, as pointed out by Weller et al. (2013). Therefore, we concluded that EBC concentrations observed at Syowa

corresponded to background EBC concentrations at the Antarctic coasts in the Indian Ocean sector.

Measurements of EBC concentrations exhibited clear seasonal features at Syowa with a maximum (median, 3.1 ng m$^{-3}$) in August–November and a minimum (1.3 ng m$^{-3}$) in January and February–April (Fig. 3). Moreover, EBC concentrations at Syowa started increasing gradually in June and July. The spring maximum mainly in October–November was observed also

at Halley, Neumayer, and the South Pole (Bodhaine, 1995; Wolff and Cachier, 1998; Weller et al., 2013). EBC measurements at Ferraz on the Antarctic Peninsula showed high EBC concentrations in September–January (Pereira et al., 2006). Therefore, similar seasonal features at all stations cannot be explained by local phenomena. Consequently, the spring EBC maximum might occur on the scale of the entire Antarctic region. Moreover, the spring EBC maximum appeared in a slightly earlier month at Syowa than in the periods examined at the other stations such as Neumayer. In addition to the spring maximum

(October–November), a second maximum of EBC concentrations was found in summer (February – March/April) at Neumayer (Weller et al., 2013) and Ferraz (Pereira et al., 2006). However, the second EBC maximum was not identified clearly at Syowa.

As pointed out by earlier studies (e.g., Wolff and Cachier, 1998; Fiebig et al., 2009; Hara et al., 2010; Weller et al., 2013), biomass burning in the middle latitudes and low latitudes has been regarded as having dominant origins of EBC measured in

the Antarctic troposphere. Biomass burning in the Southern Hemisphere occurs in Africa, South America, Australia, and Indonesia (Edwards et al., 2006a, 2006b; Ito et al., 2007; Giglio et al., 2013). The burned area in each PSA increased drastically during July–September in Africa, August–October in South America, and September–November in Australia (Giglio et al., 2013). Considering that land-origin species such as EBC and mineral dusts can outflow eastwardly to the Southern Ocean because of cyclone activity and movement (e.g., Edwards et al., 2006a; Fiebig et al., 2009; Hara et al., 2010), the contribution

of biomass burning from each PSA likely depends on where the respective coastal stations are located (e.g., sectors of Atlantic, Indian, and Pacific Oceans). For example, brief transport from southern Africa might occur rarely at Neumayer and Halley, although brief transport was observed at Syowa (Hara et al., 2010). Therefore, the difference of the month in the spring EBC maximum (August–November at Syowa, October–November at the other stations) might be associated with the seasonal variations of biomass burning in each PSA and transport pathway and processes to the Antarctic coasts. In addition to biomass

burning, anthropogenic processes (i.e., combustion of fossil fuel) must be discussed because anthropogenic EBC can outflow simultaneously from the continents in the mid-latitudes. Details of BC transport and origins will be discussed in sections 3.3 and 3.4.

## 3.2 Air mass origins at Syowa Station, Antarctica

As described above, EBC concentrations at Syowa were found to show clear seasonal variations. To elucidate the seasonal features, we must compare the seasonal features of EBC concentrations to seasonal variations of transport processes and EBC source strength in the Southern Hemisphere. Figure 4 depicts density maps of end-points of the 5-day backward trajectory (i.e. air mass origins in this study). The transport pathway is classifiable roughly into (1) poleward flow from the Southern Ocean, (2) westward flow along the coastline, and (3) outflow from the high-latitudinal Antarctic continent to the coasts. These flow patterns were identified through the year at Syowa. In the poleward flow from the ocean, air masses were transported mostly from the Atlantic Ocean of >40°S and from the Indian Ocean of >50°S within 5 days, although transport from the Atlantic Ocean at ca. 30°S was identified in some cases. The poleward flow patterns were important through the year. In addition, the poleward flow patterns were associated with eastward cyclone approach off Syowa (Hara et al., 2010). These flow patterns corresponded to EBC transport pathway to the Antarctic coasts suggested by Fiebig et al. (2009) and Hara et al. (2010). In westward flows, the air mass origins for the prior 5 days were distributed to the Antarctic coasts of approx. 150°E. Although some air masses at 150°E were transported even in winter, the density around the coasts at 140–150°E was higher in summer, particularly during November–January. Wintering research stations such as Mawson (67°36′S, 62°52′E), Zhongshang (69°22′S, 76°22′E), and Dumont d'Urville (66°40′S, 140°00′E) operated at the coasts of 40–150°E. Although EBC can be emitted from these stations by human activity, Syowa is too distant to have any strong effect on EBC concentrations by local EBC emission in these stations, as suggested by Hagler et al. (2008), because of the slight EBC source strength. Furthermore, outflows from the Antarctic continent were observed throughout the year. The trajectory density on the Antarctic continent was especially lower at high latitudes during summer. Outflow from the high-latitudinal Antarctic continent was observed under conditions with clear sky and weak winds at Syowa resulting from anticyclone influence, as described by Hara et al. (2011, 2013). By contrast, the air mass origins were distributed extensively in the Southern Oceans and Antarctic continent during winter. This difference implies that the transport strength of the outflow from the Antarctic continent had remarkable seasonal change in addition to important contribution of the poleward flow patterns from the ocean. Furthermore, air masses came occasionally from the Pacific Ocean sector across the Antarctic continent.

To understand the spatial and vertical motion of air masses, Fig. 5 shows vertical density plots of the trajectory. In the cases of poleward flow from the ocean, air masses passed through the lower troposphere, mainly in the MBL and partly in the lower free troposphere (LFT). By contrast, air masses came mostly from the free troposphere (FT) over the Antarctic continent. In addition to the descent flow, air masses near the surface on the continent were also transported to Syowa during winter. Most end-points of the backward trajectories over the continent were distributed up to ca. 4000 m during summer. However, the distribution of the end-points was expanded to ca. 6000 m over the continent during winter. It is noteworthy that the vertical density maps are shown using the height above ground level. Considering tropopause height (8–10 km) identified by $O_3$ profiles in the Antarctica during the winter (Tomikawa et al., 2009), the air mass history implies that air masses near tropopause over the continent can flow to the boundary layer (BL) at the Antarctic coasts during winter. This seasonal difference indicates that vertical mixing in the outflow from the Antarctic continent was stronger in the winter than in summer. At latitudes of around 70°S, high density was identified below 2000–3000 m. Therefore, air masses in the westward flow passed through the lower troposphere during the summer. With suggestion of vertical motion and geographical classification of air mass origins as described above, the following transport patterns and air mass origins at Syowa are finally classifiable in this study: (1) poleward flow from MBL, (2) poleward flow from LFT, (3) westward flow along the coastal line via BL, (4) westward flow along the coastal line from LFT, (5) outflow from the FT over the Antarctic continent, and (6) outflow from BL over the Antarctic continent.

As described above, EBC is expected to be supplied mostly from outside of Antarctica. Plausible transport pathways are transport via MBL and FT. We must know the EBC concentrations of each air mass origin (MBL, coastal BL, continental BL, continental FT, coastal FT, and MFT) to elucidate EBC transport pathway to the Antarctica. Figure 6 depicts seasonal features of air mass origins in each month and monthly mean and median EBC concentrations at Syowa during our measurements (2005–2016). The dominant air mass origins were MBL, coastal BL, coastal FT, and continental FT. The most dominant air mass origins were MBL and coastal BL in November–February. In addition to MBL and coastal BL, the contributions of transport from coastal FT and continental FT increased in February/March – October at Syowa, although year-to-year differences were found in the seasonal variations of air mass origins. Particularly, the contribution of transport from continental FT in March–October was higher than that in other years. This change corresponded to lower EBC concentrations in July of 2010–2012, as described above. Therefore, the increasing trend of EBC concentrations in July of 2010–2016 might not be a long-term trend but a temporal trend resulting from year-to-year variations of air mass history.

**3.3 EBC concentrations in respective air mass origins**

We compared the EBC concentrations at Syowa with the air mass history (origins) to elucidate EBC transport processes to the Antarctic coasts and EBC spatial distribution in Antarctica. The backward trajectories were computed every hour. For comparison between EBC concentrations and air mass origins, hourly mean EBC concentrations were estimated. Then, hourly EBC data were classified into each air mass origin. The hourly mean EBC concentration for each air mass origin is presented in Fig. 7. The respective EBC concentrations in MBL and marine FT were higher than those in continental FT and BL. The differences of EBC concentrations for each air mass origin might reflect on the latitudinal gradient and spatial distribution of EBC. This latitudinal gradient was found to be consistent with results reported from earlier works (Hansen et al., 1988; Bodhaine, 1995; Wolff and Cachier, 1998; Weller et al., 2013). Because of the lower (negligible) EBC source strength on the Antarctic continent, latitudinal distributions might result from dilution during transport from low latitudes and mid-latitudes and dry/wet deposition of EBC onto the snow surface.

Measurements show that EBC concentrations in each air mass origin were higher in September–November. Particularly, EBC concentrations in MBL increased gradually during May–June. Seasonal features of EBC concentrations in MBL (Fig. 7a) might correspond to those in the MBL in the Southern Ocean in Atlantic and Indian sectors, considering that air masses were transported dominantly via MBL from the mid-latitudes by the cyclone approach. Although few EBC measurements were made through the year in the Southern Ocean, EBC concentrations in Amsterdam Islands (mid-latitude in Indian Ocean: 37°50′S, 77°30′E) showed strong seasonal variations of EBC concentrations with a maximum in July–September (Wolf and Cachier, 1998; Scaire et al., 2009). Previous ship-borne EBC measurements showed EBC concentrations of <10 ng m$^{-3}$ in January–April over the southern Indian Ocean (<56°S) (Moorthy et al., 2005), and 20–80 ng m$^{-3}$ in October–December over the Indian Ocean – Southern Ocean (34–59°S) (Sakerin et al., 2007). In MBL of the southern Atlantic Ocean (close to South America) to the Southern Ocean, the EBC concentrations were <10–160 ng m$^{-3}$ in October–November and <10–120 ng m$^{-3}$ in February–March (Evangelista et al., 2007). Although we must consider the geographical locations of these EBC measurements, these EBC concentrations were several times to ten times higher than the background EBC concentrations at the Antarctic coasts, and corresponded to higher EBC concentrations at Syowa in the cases of poleward transport via MBL and MFT.

Although high EBC concentrations were obtained in air masses from MBL, we must consider EBC origins in air masses from MBL. Additionally, 120-hr backward trajectory analysis was too short to reach to contributable PSA because it took longer than one week for transport from the coasts of South America and southern Africa to Syowa (Hara et al., 2010). Density of marine traffic (i.e. ship operation) in the Southern Ocean and near the Antarctic coasts was too low to engender an increase of EBC concentrations in air mass from MBL, although ship emissions can have an influence locally on EBC concentrations, for

example ship-borne tourism in the Antarctic Peninsula during summer. Furthermore, seasonal variations and distributions of CO concentrations in the Southern Hemisphere exhibited the spring maximum corresponding to outflow from the continents and fire counts in each PSA (Gros et al., 1999; Edwards et al., 2006a, 2006b). In addition, Edwards et al. (2006a) reported that high CO concentrations in Africa appeared in earlier months than those in other PSAs such as South America. Seasonal variations of CO concentrations and fire counts (Gros et al., 1999; Edwards et al., 2006a, 2006b) were similar to the seasonal features of EBC concentrations at Syowa, as described above. Considering the highest EBC concentrations in MBL, transport via MBL from PSA with biomass burning contributed significantly to EBC concentrations in the Antarctic coasts. Similarly to CO outflow from the continents (Gros et al., 1999; Edwards et al., 2006a, 2006b), EBC outflowed from the PSA. It was subsequently transported to Antarctica, involved with cyclone activity in the Southern Ocean (Fiebig et al., 2009; Hara et al., 2010). The end-points of the backward trajectory were distributed in the lower troposphere in the marine sector (<66°S), as depicted in Fig. 5. Therefore, EBC transport to the Antarctic coasts via the free troposphere might occur at altitudes lower than 3000 m.

In spite of lower EBC concentrations in the continental FT, seasonal features of EBC concentrations reached a maximum in October–November in the continental FT (Fig. 7f), in contrast to decreased EBC concentrations in MBL and MFT from November. Additionally, the EBC concentrations from the continental FT were higher than the EBC concentrations measured at the South Pole (Hansen et al., 1988; Wolff and Cachier, 1998). Similar to the latitudinal gradient, the difference in EBC concentrations is expected to be related to the vertical gradient of EBC concentrations over the Antarctic plateau. Indeed, Schwartz et al. (2013) reported higher EBC concentrations in the upper free troposphere than in the lower troposphere over the Antarctic coasts. This fact implies that EBC was supplied to the Antarctic region also via the upper free troposphere. Figure 6 shows that no direct flow from the upper free troposphere over the marine sector to Syowa was identified. Therefore, EBC in the Antarctic coasts might be supplied also by transport via the upper free troposphere from mid-latitudes with subsequent downward flow from the continental FT. Considering the EBC concentrations in the continental FT, EBC transport via the upper FT might make a small contribution on EBC concentrations at Syowa.

## 3.4 Origins and potential source areas of EBC in the Antarctic coast (Syowa)

Trajectory analysis can provide important information about the relation between EBC concentrations and air mass history, but it cannot let us know the origins and PSA of EBC measured at Syowa (Antarctic coasts). To elucidate the BC origins and PSA, we can compare the EBC data to EBC concentrations simulated using the CHASER model. Figure 8 presents seasonal features of monthly median EBC concentrations measured at Syowa and the model-simulated BC (mBC) concentrations. The mBC concentrations tended to be lower than the EBC concentrations in the summer, although the mBC concentrations were higher during the spring maximum. This difference might result from positive bias using filter-based BC measurement techniques such as the use of aethalometer (e.g., Bond et al., 2013) and uncertainty of EBC transport strength to the Antarctic region involved with aging and deposition processes in the model simulation. Furthermore, the range of the mBC concentrations and their seasonal features were consistent with those of the observed data (median EBC concentrations): $[mBC] = 0.935 \times [EBC]_{observed} - 0.0588$ ($R$=0.5771). Therefore, we discuss EBC origins and PSA using the model data presented below.

In this study, the following EBC origins were classified: (1) biomass burning (BB) such as forest and savanna fires, (2) fossil fuel combustion (FFC), and (3) other combustion (OC). Because "other combustion" includes combustion of biomaterials (e.g. wood fuels), most of the other combustion data were those from combustion of biomass in a broad sense. Contributions of potential EBC origins showed clear seasonal features as presented in Fig. 9a. Biomass burning was dominant (50–80%, mean 70.7%) in spring EBC maximum at Syowa. By contrast, the FFC contribution was lower (10–20%, mean 14.8%) than the BB

contribution. As described above, earlier results of studies have shown that EBC in the Antarctic troposphere was supplied by BB in the Southern Hemisphere and long-range transport (Wolff and Cachier, 1998; Fiebig et al., 2009; Hara et al., 2010; Weller et al., 2013). Although the OC contribution increased to more than 50% in autumn–winter (February–June), the periods corresponded to the lower EBC concentrations at Syowa.

Figure 9b shows that the BB-mBC concentrations reached their maximum values at Syowa during August–October, although high BB-mBC concentrations were found occasionally in July and November. South America, southern Africa, and Australia were identified as the important PSAs of BC at Syowa (Fig. S5 in Supplementary Materials). Particularly, BB from South America and southern Africa contributed more than 90% of BB-mBC in the spring maximum (Fig. S4). The contributions of BB in South America and southern Africa in August–November were, respectively, 18.1–62.3% (mean 42.1%) and 15.9–71.7% (mean 43.3%). Relative importance of BB in South America and southern Africa showed a slight year-to-year difference. Moreover, BB-mBC concentrations in southern Africa increased often in earlier months than they did in South America. The BB-mBC concentrations in Australia increased in November (after the spring maximum) at Syowa. Contributions and concentrations of BB-mBC in Australia increased drastically after the spring maximum (October–November). However, BB-mBC concentrations in Australia were considerably lower at Syowa than those in southern Africa and South America. These differences of seasonal features of BB-mBC concentrations/contribution at Syowa in each PSA might be associated with (1) the seasonality of occurrence of BB in each PSA and (2) transport strength from each PSA to Syowa. Indeed, earlier works showed similar seasonal features of fire counts, burned areas, and CO concentrations/emission in/over South America, southern Africa, and Australia (Edwards et al., 2006a, 2006b; van der Werf et al., 2006; Giglio et al., 2013). Furthermore, seasonal features of fire counts and aerosol absorption optical depth measured by satellite (OMI) showed a one-month lag in the maximum of aerosol absorption optical depth (September for Africa and October for South America; Torres et al., 2010). In addition, mineral dust and atmospheric substances from biomass burning can outflow eastwardly from the continents (Edwards et al., 2006a; Fiebig et al., 2009; Hara et al., 2010; Neff and Bertler, 2015). The eastward flow patterns in the Southern Hemisphere are presented in Fig. 4, suggesting strongly that BB-mBC from southern Africa and South America can be transported directly to Syowa, but rarely from Australia. Consequently, BB-mBC from Australia might usually have a lower contribution at Syowa.

Model simulation showed high BB-mBC concentrations in Australia in later spring of 2011 and 2012. Higher EBC concentrations were observed in November 2011 than in November in other years. We must ascertain the transport pathway from Australia to the Syowa to understand the high BB-mBC concentrations in Australia. As shown by Neff and Bertler (2015), air masses extended from Australia to the Antarctic coasts in the Pacific Ocean sector. For that reason, BB-mBC originated in Australia can be transported to Syowa, considering the westward flow along the coastline, as described above. Similarly, BB-mBC in southern Africa might contribute only slightly to EBC observed at Halley and Neumayer because of rarely directed transport. Therefore, a lag in the spring maximum of EBC concentrations among Syowa, Halley and Neumayer might be attributed to the seasonality of BB phenomena in each PSA and the transport strength from PSA to each station.

The BB-mBC concentrations at Syowa started increasing in June (Fig. 9b). They increased considerably in August. The EBC concentrations increased gradually after May–June. In contrast to BB-mBC, the concentrations of FFC-mBC and OC-mBC started increasing in May–June (Fig. 9c). Furthermore, good correlation was found between the concentrations of FFC-mBC and OC-mBC ($R^2$=0.9675), with lower correlation in BB-mBC–FFC-mBC ($R^2$ = 0.4175) and BB-mBC–OC-mBC ($R^2$ = 0.3654). The good correlation found between FFC-mBC and OC-mBC implies strongly that seasonal features of FFC-mBC and OC-mBC might reflect variations of transport strength from each PSA to Syowa.

South America was found to be the most-contributing PSA (34.1–82.4%; mean, 63.6%) in FFC-mBC at Syowa through the year (Figs. 9c, S5). The FFC-mBC contribution in southern Africa was 7.4–54.0% (mean, 20.9%). In spite of the larger BB contribution in Australia, FFC had a contribution of only 3.9–17.5% (mean, 8.0%), which was lower than that of BB-mBC in Australia. The contributions of FFC and OC differed greatly from those of BB, especially in South America and southern Africa. The relevant likelihoods must be discussed to elucidate this difference: (1) difference of transport pathway of anthropogenic EBC from South America and southern Africa to the Antarctica and (2) differences of EBC emission from anthropogenic combustion (i.e. fossil fuel use) in South America and southern Africa. Because of eastward cyclone movement in the Southern Ocean, air masses outflowed eastwardly from the continents of South America and southern Africa. Unlike the Africa continent, the South American continent extends to ca. 55°S. This geographical difference can engender higher contributions of anthropogenic EBC emitted from South America. Indeed, direct evidence of EBC transport from South America was reported in earlier works (Pereira et al., 2006; Fiebig et al., 2009; Hara et al., 2010). In addition, higher contributions of South America were observed in transport of mineral dusts to the Antarctica (e.g., Delmonte et al., 2004, 2008; Gassó et al., 2010; Li et al., 2010). Considering that both BB-mBC and FFC-mBC outflowed simultaneously from each PSA, the transport processes cannot account for the difference between contributions of BB-mBC and FFC-mBC. Unlike FFC, BB has strong seasonality in each PSA as described above. Furthermore, fossil fuel consumption depends on the gross domestic product (GDP). Indeed, the aggregated GDPs of countries in South America are the largest in the Southern Hemisphere. Therefore, the difference of FFC-mBC contribution might reflect the fossil fuel consumption related to population and economic activity in the respective PSAs. Because of rapid population expansion in countries of southern Africa recently, it is expected that more EBC can be released by fossil fuel combustion in southern Africa in the future. Therefore, continual EBC measurements must be conducted at the Antarctic coasts to monitor the atmospheric substances (e.g. EBC) originating from combustion in the Southern Hemisphere.

The concentrations of BB-mBC, FFC-mBC, and OC-mBC showed minima during February–April/May, although poleward flow via MBL and MFT occurred as portrayed in Fig. 6. The following possibilities are contributing factors: seasonal features of (1) EBC source strength in each PSA and (2) air mass history (i.e. air mass origins). As demonstrated by earlier work (Edwards et al., 2006a, 2006b; van den Werf et al., 2006; Torres et al., 2010), EBC emissions from BB in South America, southern Africa, and Australia showed strong seasonal variation, with lower fire counts in February–April/May because of the large precipitation amounts. In the CHASER model, fresh BC immediately after release from BB is assumed as hydrophobic, so that few fresh BC might be removed by precipitation near source areas. Aging processes during transport can engender gradual change into internal mixtures of BC with a hydrophilic surface. Then, BC can be scavenged through wet deposition during transport. Considering clear seasonal features of CO with longer residence time than BC (Edwards et al., 2006a, 2006b; van den Werf et al., 2006), seasonal variation of BC emissions might have a greater contribution to seasonal features of mBC and EBC at Syowa than wet deposition of BC during transport.

Results show that BB was the greatest factor affecting EBC concentrations in the Southern Hemisphere. Therefore, the seasonal features of biomass burning might affect EBC concentrations in the Antarctic region. However, it is noteworthy that FFC-mBC concentrations also showed a minimum in February–April/May at Syowa. This fact cannot be explained by the seasonal features of BB. Assuming that FFC-mBC is useful as a proxy for transport from the continents with human activity to the Antarctic region, the EBC minimum in February–April/May might be associated not only with the features of BB, but also with the features of the air mass history. As portrayed in Fig. 6, the contributions of air masses identified as coastal BL and continental FT increased in February–April/May in the most years, and particularly in 2010. A similar tendency (high contribution of the upper atmosphere at surface of Antarctic coasts in austral autumn) was also identified from FLEXPART analysis by Stohl and Sodemann (2010). The EBC source strength in the Antarctic region, especially in FT over the Antarctic

continent, is less or negligible. Therefore, the seasonal features of BB in the Southern Hemisphere and air mass origins at Syowa might engender the EBC minimum at Syowa.

## 3.5 Variations of AAE at Syowa Station, Antarctica

In addition to seasonal features of EBC concentrations (Fig. 3), AAE showed clear seasonal variation at Syowa (Fig. 10). AAE ranged mostly in 0.5–1.0 in April–October and 1.0–1.5 in summer (December–February) during our measurement at Syowa. AAE increased slightly in the maximum of EBC concentrations (September–October). Earlier works confirmed AAE = 1 in the case of dominance of external mixed EBC (e.g., Bond et al., 2013; references therein). By contrast, AAE of the coated BC (i.e. internal mixed EBC) is lower than 1 (Leck and Cappa, 2010; Bond et al., 2013). Aethalometer measurements cannot provide direct information related to the mixing states of carbonaceous aerosols. As suggested by Fiebig et al. (2009) and Hara et al. (2010), EBC is expected to be transported from mid-latitudes and low latitudes. Therefore, EBC mixing states might be changed by aging processes near source regions and during long-range transport (Shiraiwa et al., 2007; Saleh et al., 2013, 2014; Ueda et al., 2018). Indeed, Ueda et al. (2018) showed that EBC was present mostly as internal mixtures in the marine boundary layer (MBL) of the Southern Ocean. The CHASER model also indicates that internal mixing states of BC were dominated through the year (not shown, details published elsewhere). Therefore, lower AAE in April–October might result from the dominant presence of coated EBC particles (internal mixtures) at Syowa. The slight AAE increase corresponded to the spring maximum of EBC concentrations. The following possibilities were considered for the slight AAE increase: (1) change of mixing states of BC and (2) contribution of other light-absorbing materials such as organic aerosols and minerals. The organic aerosols and minerals have high AAE, for instance, 3.5–7 for organics and typically 2–3 for minerals (e.g., Bond et al., 2013, and references therein). Although the internal mixing states of BC were dominant in CHASER model simulation, external mixtures of BC increased in spring EBC maximum (not shown, details published elsewhere). Considering AAE of external mixing of BC, increase of external mixing of BC can engender an AAE increase. Additionally, spring EBC maximum at the Antarctic coasts was associated closely with biomass burning. Organic aerosols with high AAE derived from biomass burning were expected to be transported simultaneously into Antarctica. Consequently, transport of organic aerosols might contribute to the slight AAE increase in September – October. By contrast, high AAE in summer cannot be explained solely by EBC aging processes. Earlier results of studies have shown high concentrations of mineral particles (Al) at Neumayer in the summer (Wagenbach, 1996; Weller et al., 2013). Therefore, it is necessary to assess the effects of organic aerosols and mineral particles on optical absorption during summer.

Considering strong optical absorption by organics (i.e. BrC) in the UV range, contribution of organics can be assessed from comparison between AAE in UV-IR channels (AAE$_{UV-IR}$, λ=370–950 nm) and AAE in Vis-IR channels (AAE$_{Vis-IR}$, λ=590–950 nm). Figure 11 depicts the relation between AAE$_{UV-IR}$ and AAE$_{Vis-IR}$. Correlation of AAE$_{UV-IR}$ and AAE$_{Vis-IR}$ was observed throughout the year. Particularly, high correlation ($R^2$ > 0.7) was obtained in March and September–December. Monthly median ratios of AAE$_{Vis-IR}$/AAE$_{UV-IR}$ were 0.55–0.92 (Fig. 11b). Particularly, higher ratios (0.88–0.92) were found in the spring maximum of the EBC concentration (September–November). High optical absorption by organic aerosols in the UV ranges engenders an increase of AAE$_{UV-IR}$ and the higher ratios (e.g., Bond et al., 2013). Therefore, the difference suggests that organic aerosols, rather than effects of mineral particles, contributed to optical absorption and AAE. Results of linear regression, as portrayed in Fig. 11, indicated that intercepts had negative values in all months. Particularly, larger negative intercept values (≤ -0.4) were obtained for October–November (Fig. 11d). Considering that EBC in EBC maximum might be associated with biomass burning and long-range transport (details discussed in later sections), it is expected that large amounts of biomass-burning-origin organic aerosols are transported to Antarctica. Indeed, the concentrations of particulate oxalate show a spring maximum (Fig. S6 in Supplementary Materials). Additionally, high concentrations of oxalate and brown carbons are associated

with secondary organic aerosol formation in a condensed phase (e.g., Zhang et al., 2012). When optical absorption in UV regions was increased by organic aerosols (i.e. BrC), correlation between $AAE_{UV-IR}$ and $AAE_{Vis-IR}$ can be shifted to larger $AAE_{UV-IR}$ region. This change might engender larger negative intercept values. Therefore, the larger negative intercepts in October–November might result from effects of organic aerosols derived from biomass burning.

Considering that EBC concentrations decreased in summer (December–February) at Syowa, organic aerosols and their precursors might be supplied not from combustion processes but from the other sources. One major organic aerosol constituent was found to be $CH_3SO_3^-$ (Fig. S6 in Supplementary Materials), which is involved with oceanic bioactivity and photochemical processes in the Antarctic coasts during summer (e.g., Minikin et al., 1998; Preunkert et al., 2008). Indeed, $CH_3SO_3^-$ was identified as an internal mixture with $SO_4^{2-}$ in the Antarctic coasts (Hara et al., 1995). In addition, aerosol particles containing $SO_4^{2-}$ were present as strong acidic droplets in the Antarctic troposphere (e.g., Hara et al., 2013, references therein). Therefore, $CH_3SO_3^-$ might be present as an acidic solution in aerosols in the Antarctic troposphere during summer. $CH_3SO_3H$ aqueous solutions have strong optical absorption in UV (Fig. S7 in Supplementary Materials). Indeed, the imaginary refractive index has a weak band in the UV region (Myhre et al., 2004). Therefore, AAE in the summer (December–February) might be associated with EBC aging processes and with the presence and mixing of organic aerosols (e.g. $CH_3SO_3H$) derived from oceanic bioactivity.

In contrast to the AAE summer maximum, AAE increased slightly in the spring EBC maximum (Fig. 10). Moreover, slopes of relation of $AAE_{UV-IR}$-$AAE_{Vis-IR}$ in the linear regressions during the EBC maximum exhibited maximum (slopes $\geq$ 1) at Syowa, as portrayed in Fig. 11. The concentrations of EBC and organic aerosols derived from biomass burning increased in the spring maximum as described above, whereas the EBC concentrations decreased and the concentrations of organic aerosols such as $CH_3SO_3^-$ derived from oceanic bioactivity increased during summer. In addition, the optical absorption of BrC was found to vary greatly depending on the origins of BrC (Moschos et al., 2018). Therefore, these differences might engender seasonal features of aerosol optical absorption properties related to the concentrations of EBC and organic aerosols, optical absorption properties of BrC, and mixing states of aerosol constituents at Syowa.

## 4. Conclusion

EBC measurements have been conducted at Syowa Station, Antarctica since February 2005. Long-term trends over approximately a decade were almost constant in 2005–2016. Seasonal features of EBC concentrations at Syowa showed maximum values in September–October and minimum values in February–April, similar to the seasonal features observed at Neumayer, Halley, and the South Pole (Wolff and Cachier, 1998; Weller et al., 2013). Comparison between EBC concentrations and air mass origins identified by backward trajectory implies that EBC in the Antarctic coasts was supplied mostly by transport via MBL and lower free troposphere, particularly during winter–spring. Additionally, some EBC was to the burden on Antarctica through the upper troposphere; it was then transported downward from the continental free troposphere to near the surface at Syowa. The EBC minimum might be attributable to general transport patterns (higher contributions of the free troposphere and coastal boundary layer). The CHASER model simulation showed that the most important origins and PSA of EBC at Syowa Station were biomass burning in South America and southern Africa. Fossil fuel combustion in South America and southern Africa also have important contributions. South America is the most important PSA of mBC derived from fossil fuel combustion. Aerosol optical properties based on AAE showed clear seasonal features of AAE with 0.5–1.0 during April–October and a maximum (1.0–1.5) in December–February. The AAE features might be associated with organic aerosols and mixing states of EBC. With population growth and economic development in the Southern Hemisphere, more anthropogenic BC is expected to be released in the future. Moreover, the PSA of EBC is apparently different

between Neumayer in the Atlantic Ocean sector and Syowa in the Indian Ocean sector. Therefore, continual EBC measurements must be taken at Syowa Station to elucidate the EBC burden into the Antarctic region and effects on surface albedo and atmospheric aerosol absorbing in the future. Furthermore, these data can provide a better understanding and interpretation of EBC records in Antarctic ice cores from the perspectives of transport processes and the biomass burning history.

**Appendix**

**List of acronyms used in this study**

AAE       Absorption Ångström exponent

$AAE_{UV-IR}$       Absorption Ångström exponent in range of UV-IR ($\lambda$ = 370–950 nm)

$AAE_{Vis-IR}$       Absorption Ångström exponent in range of Vis-IR ($\lambda$ = 590–950 nm)

BB       Biomass burning

BC       Black carbon

BL       Boundary layer

BrC       Brown carbon

$C_{abs}$       Optical absorption cross-section

CCM       Chemistry climate model

CHASER       Chemical atmospheric global climate model for studies of atmospheric environment and radiative forcing

EBC       Equivalent black carbon. In this study, we use the term EBC for the corrected mass BC concentrations and the measured BC concentrations using filter-base optical techniques in previous works.

EC       Elemental carbon

ENSO       El Niño – Southern Oscillation

ESM       Earth System Model

FFC       Fossil fuel combustion

FT       Free troposphere

GDP       Gross domestic product

HYSPLIT       Hybrid Single-Particle Lagrangian Integrated Trajectory model

IR       Infrared

JARE       Japanese Antarctic research expedition

LFT       Lower free troposphere

MAAP       Multi-angle absorption photometer

mBC       Model BC (estimated using the CHASER model)

MBL       Marine boundary layer

MFT       Marine free troposphere

MIROC       Model for Interdisciplinary Research On Climate

NMVOCs       Non-methane volatile organic carbons

OC       Other combustion

PSA       Potential source area

PSAP       Particulate soot absorption photometer

rBC       Refractory BC

SP2       Single particle soot photometer

SPRINTARS    Spectral Radiation – Transport Model for Aerosol Species

UV    Ultraviolet

Vis    Visible

**Author contributions**

K.H., K.O., M.S., and T.Y. designed the experiments, which were conducted by K.O., K.H., and M.Y. K.H. wrote the manuscript and analyzed BC data and backward trajectory. K.H. and M.Y. analyzed and discussed aerosol optical properties. K.S. and T.O. developed and conducted the tagged BC simulation using the CHASER model. All authors reviewed and commented on the paper.

**Acknowledgments**

We would like to thank Y. Aoyama, Y. Takeda, T. Masunaga, T. Kinase, C. Ikeda, Y. Hayakawa, J. Matsushita, and I. Arakawa for help with aerosol measurements at Syowa Station, Antarctica, and C. Nishita-Hara for measurement of absorbance of MSA aqueous solution. We obtained MACC reanalysis data from the European Centre for Medium-Range Weather Forecasts (ECMWF). This study was supported by the "Observation project of global atmospheric change in the Antarctica" for JARE 43–47, and Grants-in-Aid for Scientific Research (B) (no. 22310013 and 15H02806, PI: K. Hara) from the Ministry of

Education, Culture, Sports, Science and Technology of Japan.

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

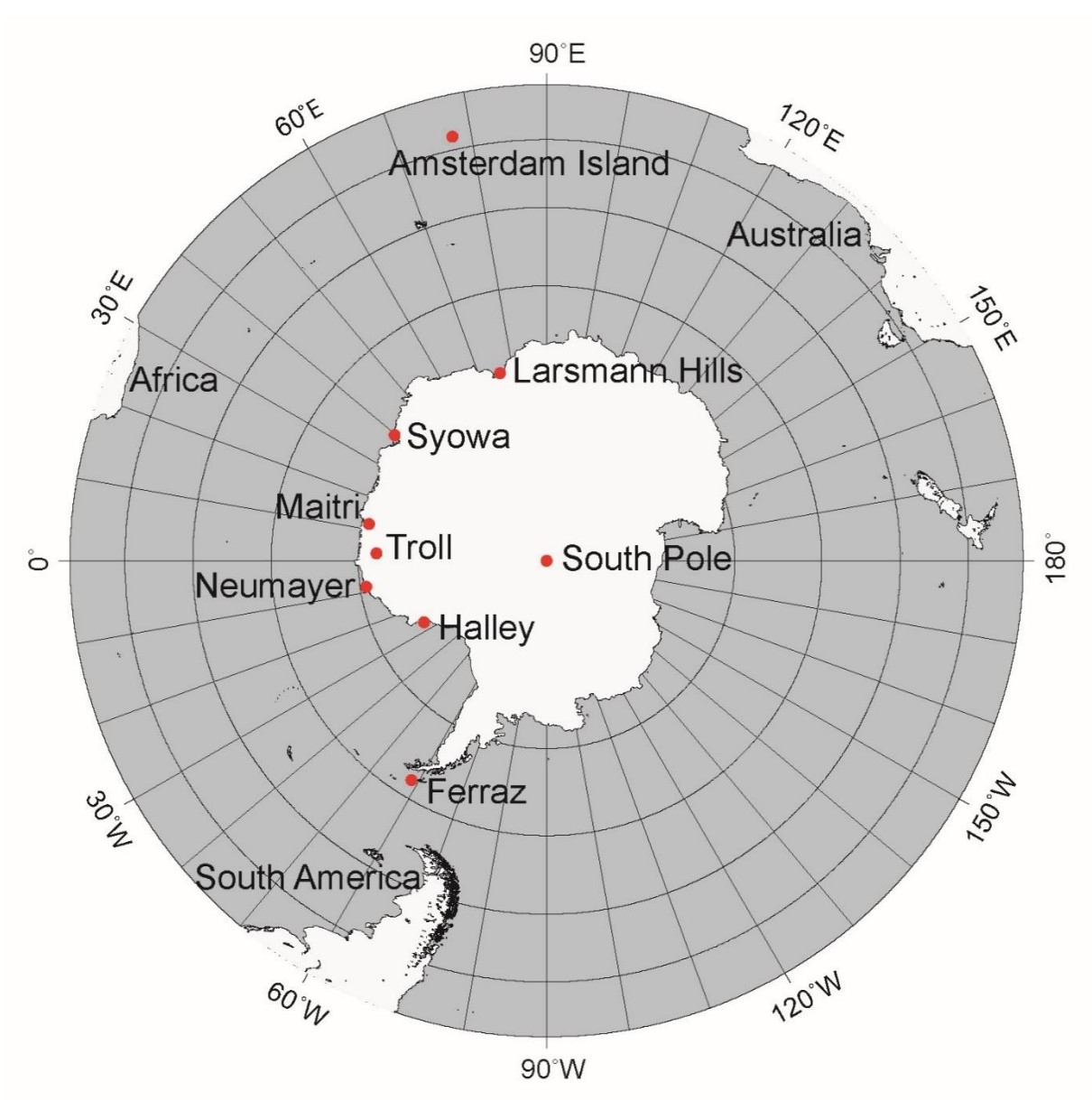

**Figure 1: Locations of Syowa Station and other research stations with BC measurements in Antarctica and the Southern Ocean.**
Red circles represent locations of each station.

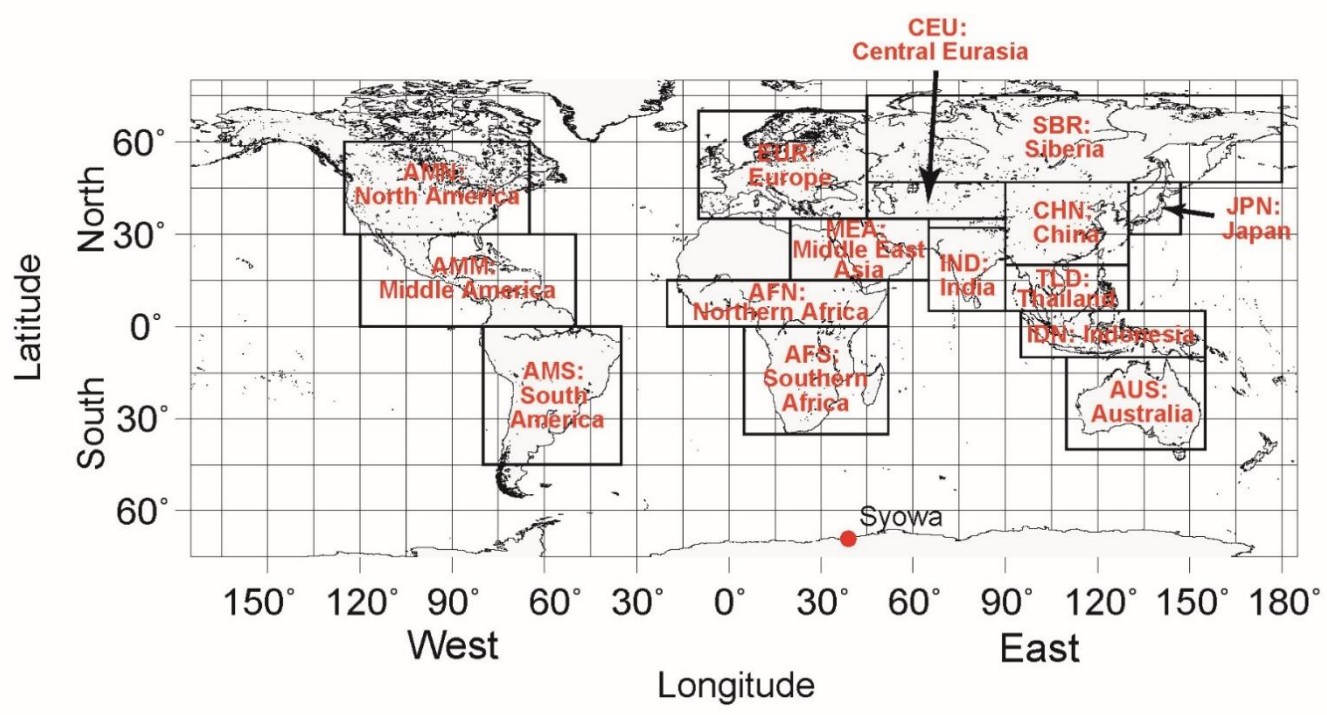

**Figure 2: Regional separation for BC tracer tagging. Red circle represents the location of Syowa Station, Antarctica.**

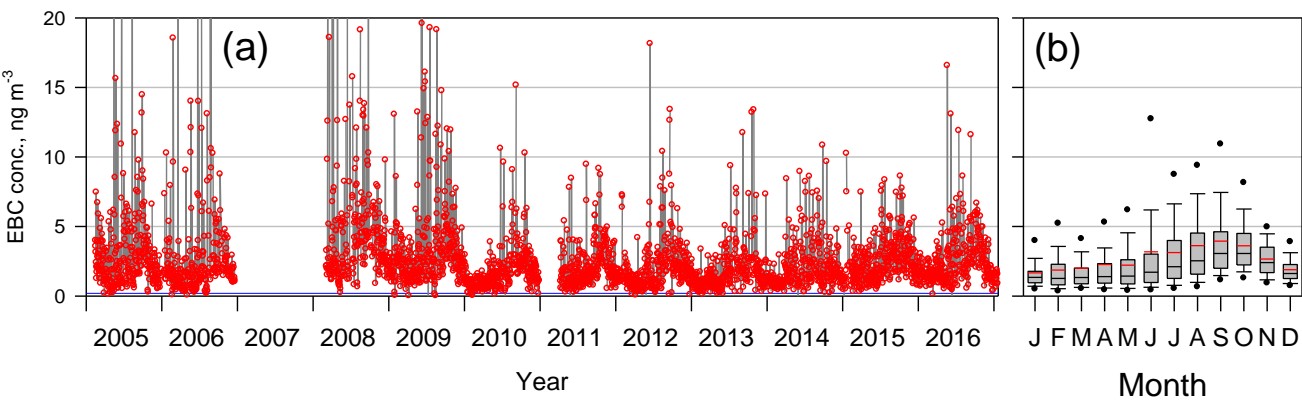

**Figure 3: Seasonal features of (a) daily median EBC concentrations and (b) monthly box plot of EBC concentrations at Syowa from February 2005. EBC concentrations were not available in Jan. 2007 – Jan. 2008 and January 2011 – early April 2011 because of mechanical troubles of the aethalometer. Blue line in (a) shows the detection limit (0.2 ng m$^{-3}$) in our measurements. In box plots, the upper bar, upper box line, black middle box line, bottom box line, and bottom bar respectively denote values of 90%, 75%, 50% (median), 25%, and 10%. The red line shows mean values.**

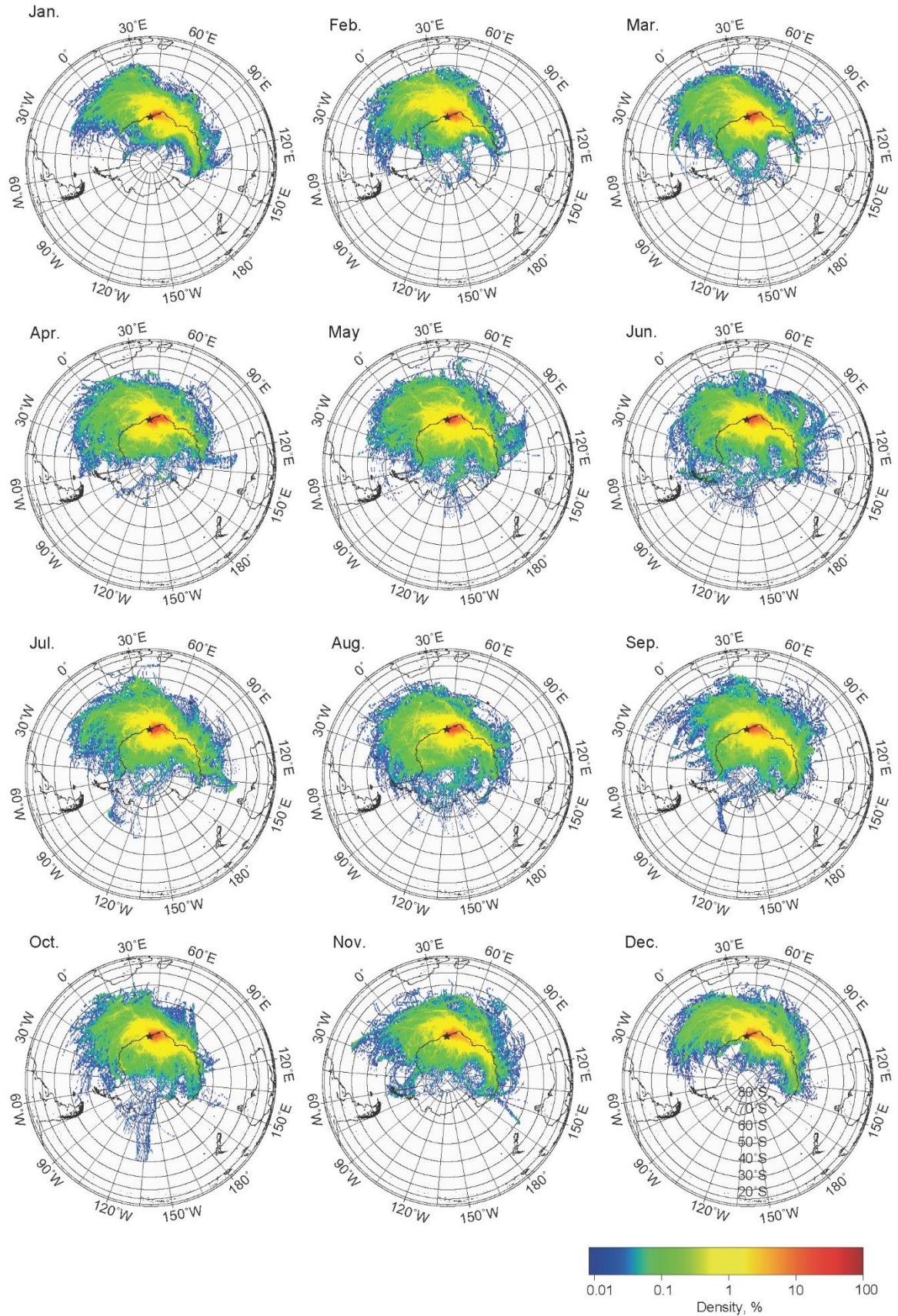

**Figure 4: Density map of air mass origins in each month at Syowa Station. Black stars denote the respective locations of Syowa Station.**

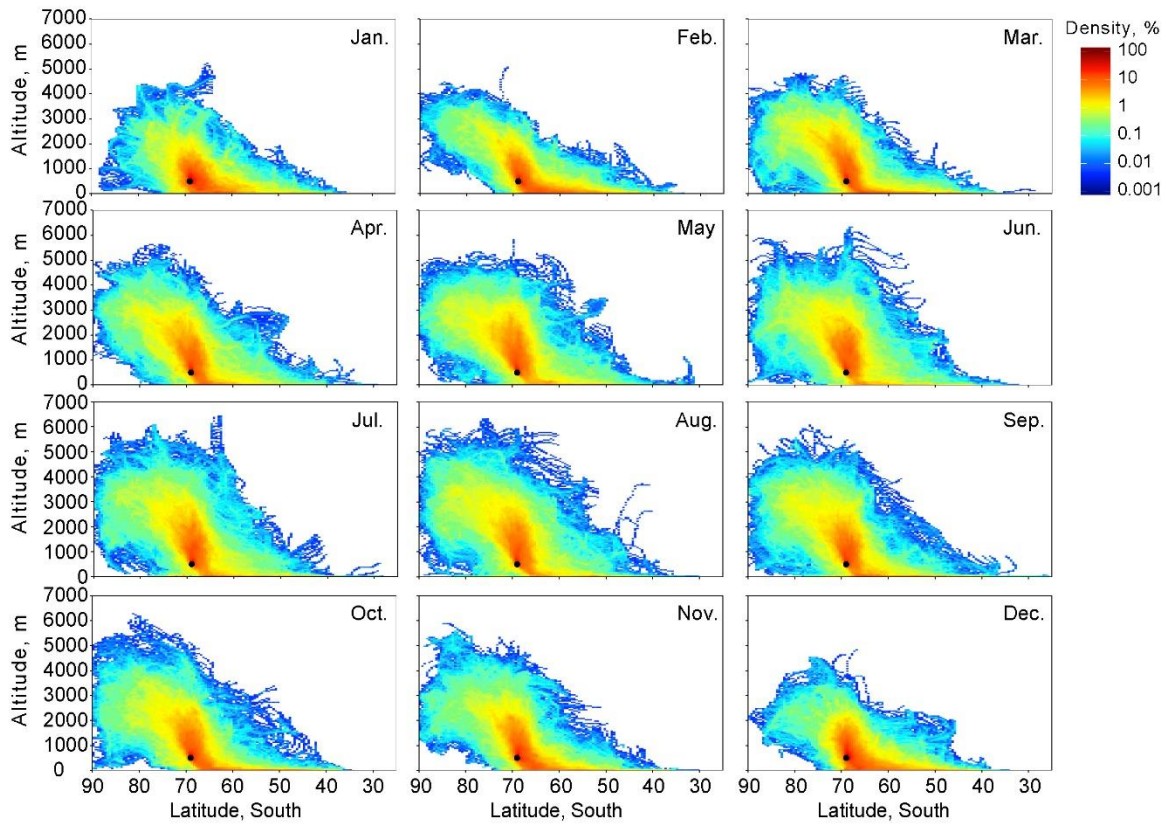

**Figure 5: Density plots of vertical motion of air masses transported to Syowa Station in each month. Black points show initial points of the trajectory analysis over Syowa Station. Altitudes are given "above ground level".**

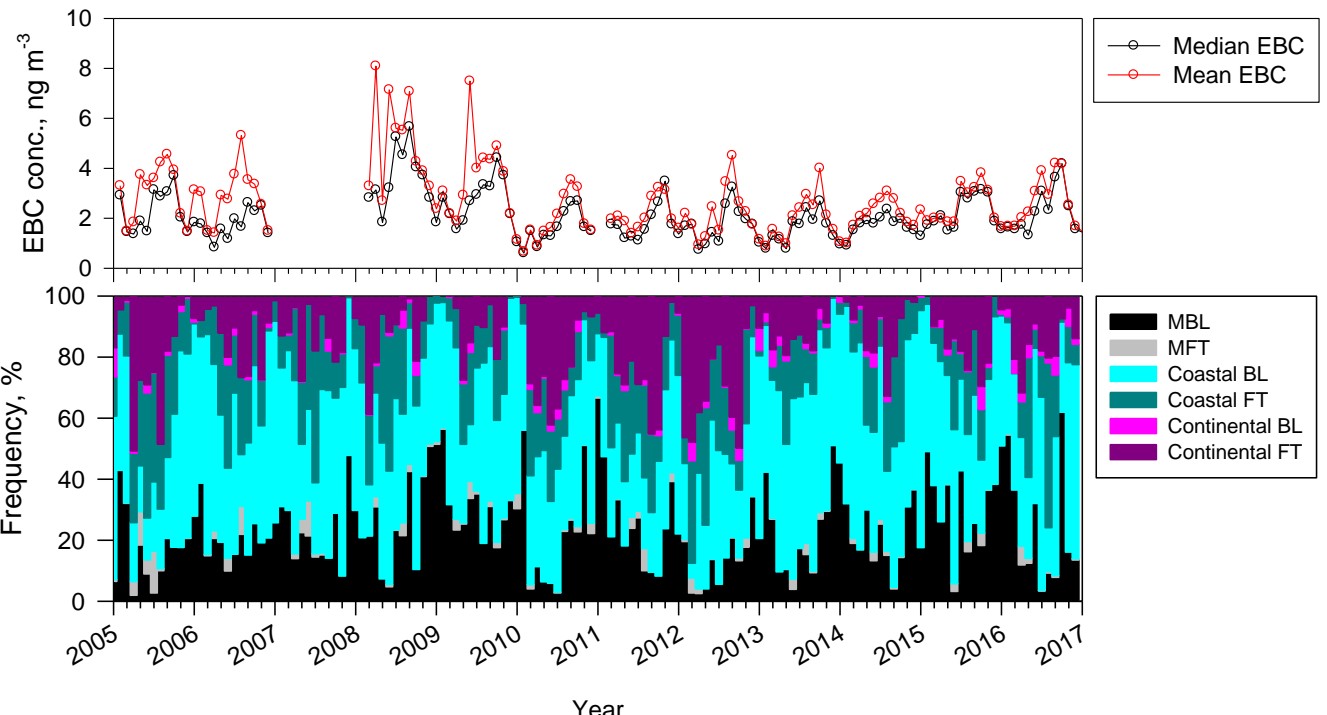

**Figure 6: Seasonal features of monthly mean and median EBC concentrations, and air mass origins at Syowa Station.**
**MBL, MFT, coastal BL, coastal FT, continental BL, and continental FT respectively denote the marine boundary layer,**
**marine free troposphere, coastal boundary layer, coastal free troposphere, continental boundary layer, and continental**
**free troposphere.**

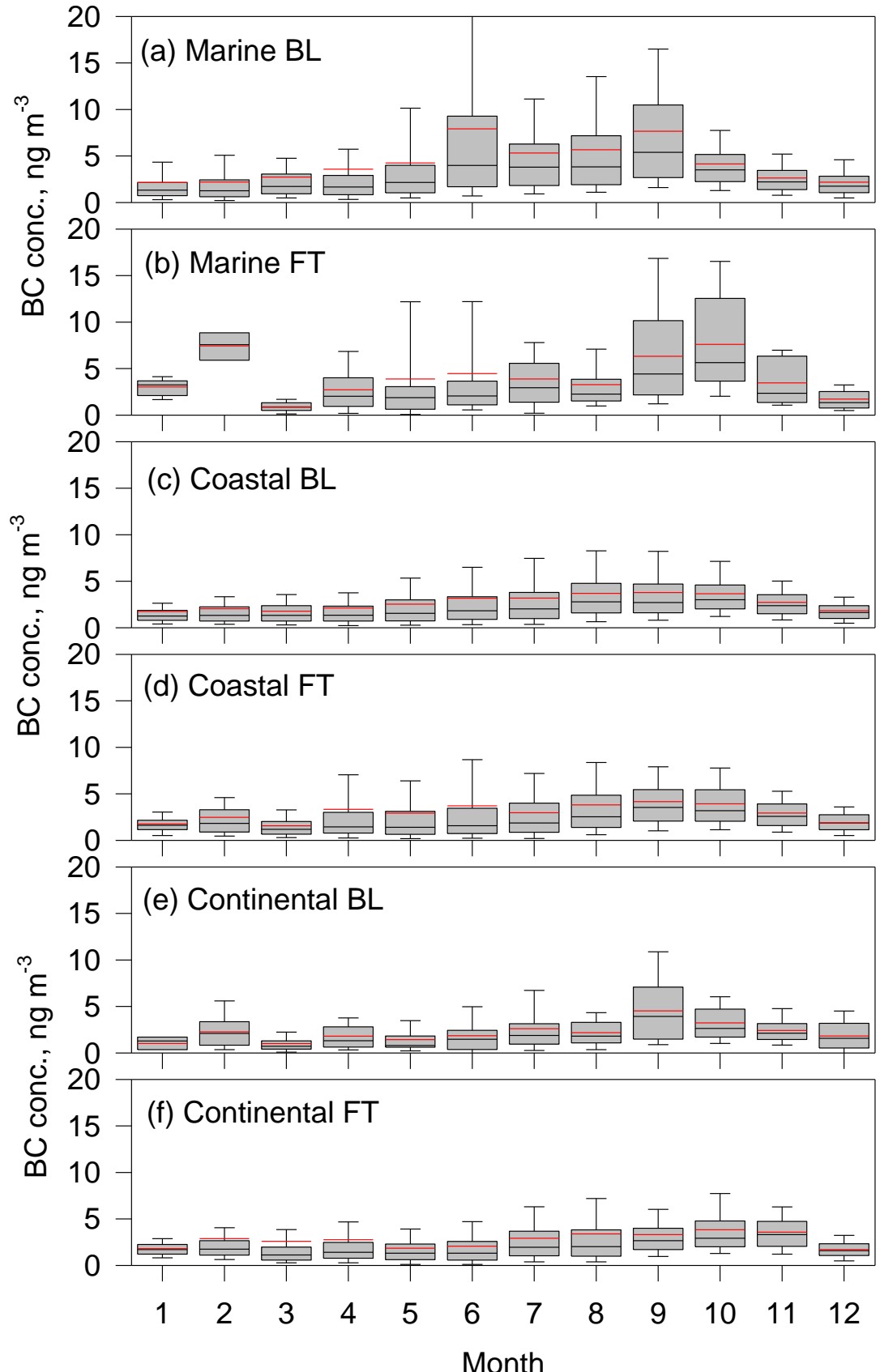

**Figure 7: Seasonal features of hourly mean EBC concentrations in each air mass origin at Syowa Station. MBL, MFT, coastal BL, coastal FT, continental BL, and continental FT respectively denote the marine boundary layer, marine free troposphere, coastal boundary layer, coastal free troposphere, continental boundary layer, and continental free troposphere. In box plots, the upper bar, upper box line, black middle box line, bottom box line, and bottom bar respectively denote values of 90%, 75%, 50% (median), 25%, and 10%. The red line represents mean values.**

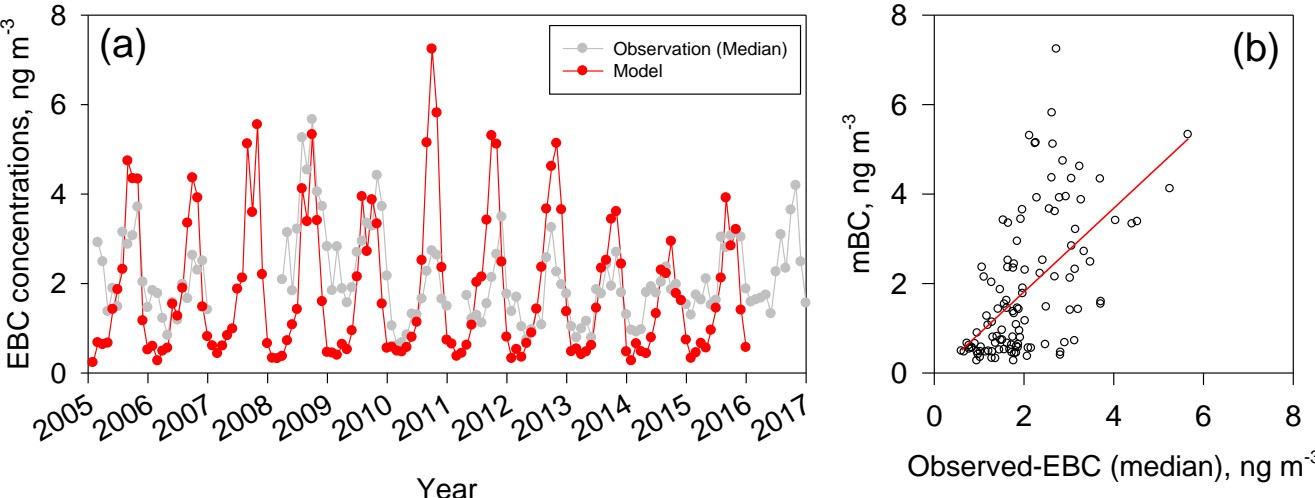

**Figure 8: (a) Seasonal features of monthly median EBC concentrations and mBC concentrations at Syowa Station and (b) the relation between monthly median EBC concentrations and mBC concentrations. The red line in (b) shows the regression line.**

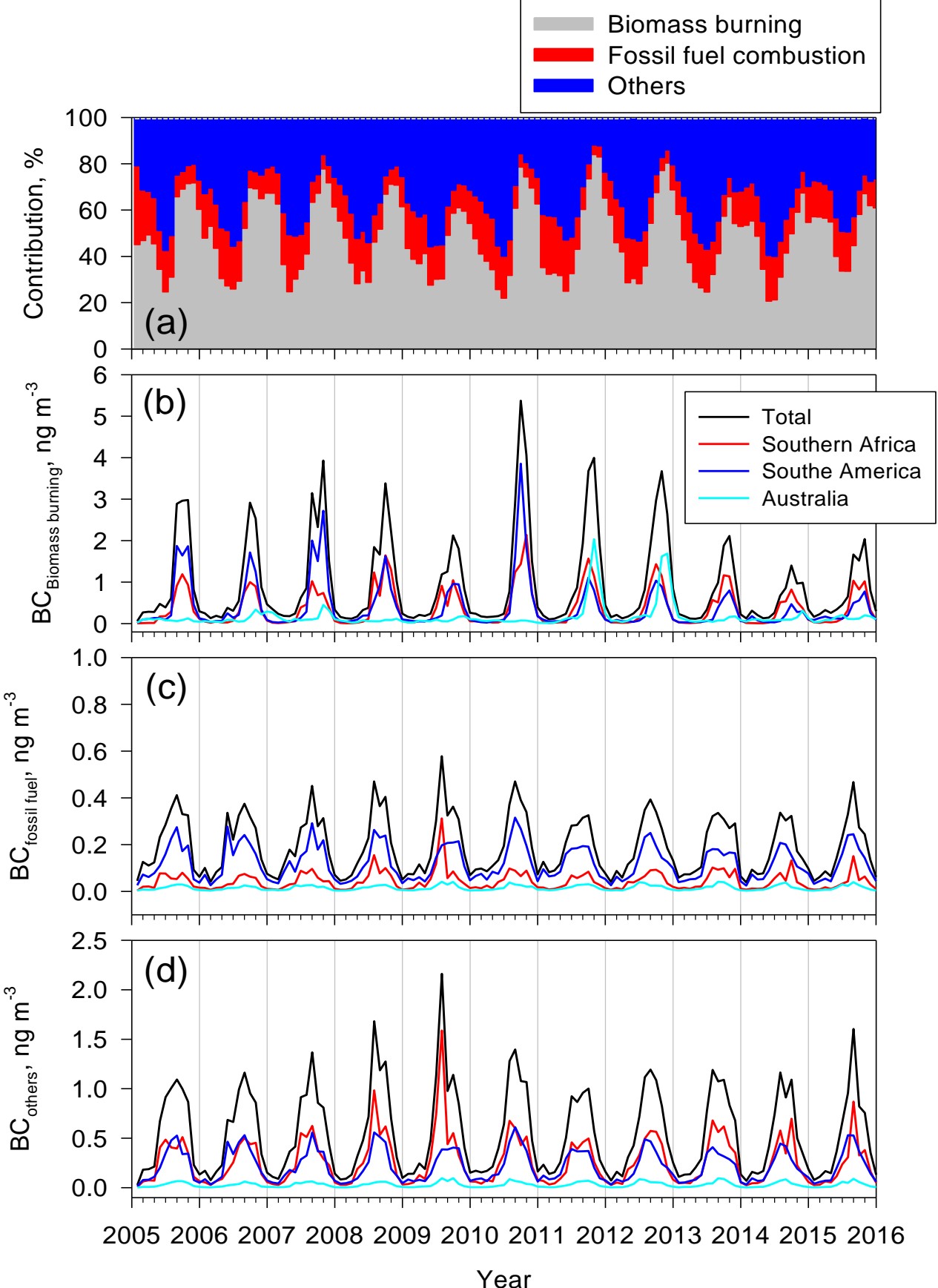

**Figure 9: Seasonal features of (a) contribution of potential origins of mBC at Syowa Station, (b) the concentrations of mBC released from biomass burning in major PSA, (c) the concentrations of mBC released from combustion of fossil fuels in major PSA, and (d) the concentrations of mBC released from the others in major PSA.**

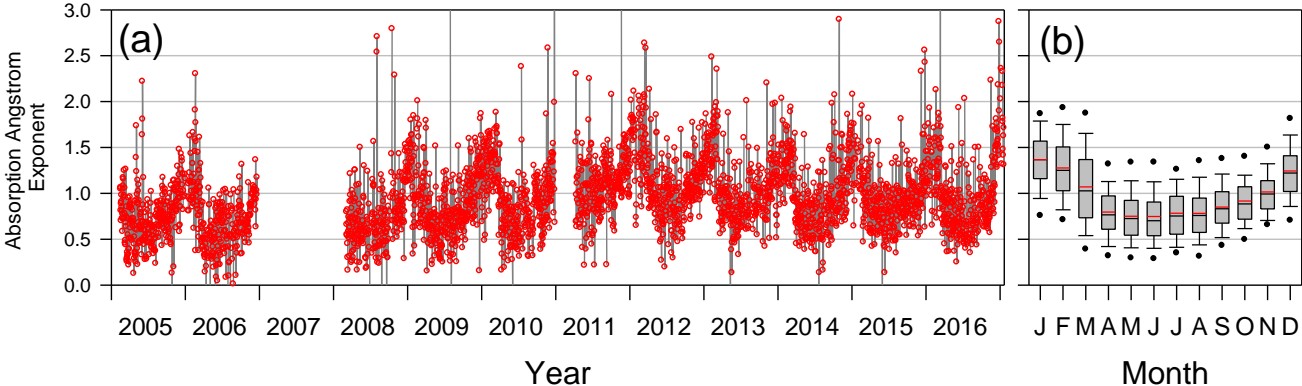

**Figure 10: Seasonal features of (a) daily median AAE (λ = 370–950 nm) and (b) monthly box plot of AAE at Syowa since February 2005. AAE data were not available in Jan. 2007 – Jan. 2008 and January 2011 – early April 2011 because of mechanical troubles in the aethalometer. In box plots, the upper bar, upper box line, black middle box line, bottom box line, and bottom bar respectively denote values of 90%, 75%, 50% (median), 25%, and 10%. The red line shows mean values.**

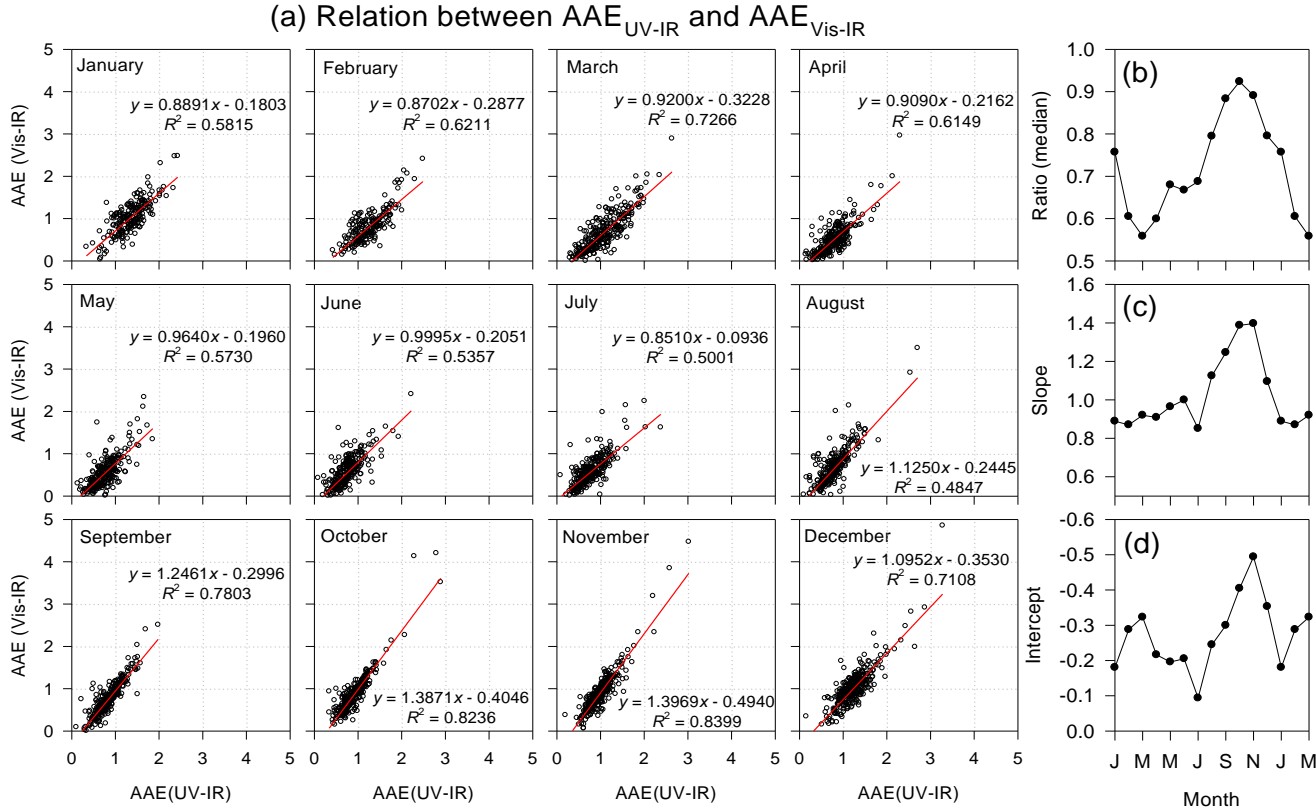

**Figure 11: (a) Relations between absorption Angstrom exponent (AAE) in UV-IR channels and AAE in Vis-IR channels, and seasonal features of (b) ratios of AAE$_{Vis-IR}$/AAE$_{UV/IR}$, and (c) slope and (d) intercepts in the linear regression lines. AAE data with BC concentrations lower than 0.2 ng m$^{-3}$ (detection limit) were excluded from the plots. Red lines represent regression lines.**