# Peer review of "Seasonal features and origins of carbonaceous aerosols at Syowa Station, coastal Antarctica"

_Atmospheric Chemistry and Physics, 2018_

## Referee Comment (RC1) · Anonymous Referee #1 · 30 Jan 2019

This work describes 11 years of equivalent black carbon (eBC) measurements at Syowa Station in Antarctica and contributes to our understanding of eBC source regions, optical properties and its seasonal cycle in Antartica. This work represents a set of measurements that are crucial to understanding the composition of the remote atmosphere and is therefore deserving of publication in Atmospheric Chemistry and Physics. Following are suggestions to improve the paper.

General Comments:

(1) This paper requires a thorough edit for clarity. In particular there is a tendency toward paragraphs that focus on multiple ideas and that lack clear topic sentences or conclusions. An examination of the order in which ideas are presented may also be helpful.

[Figure]

(2) This paper make some inferences based on indirect observations, which are identified specifically below. I suggest that the authors ensure that their stated conclusions follow directly from the data they have presented.

(3) The text would benefit from more clear and concise explanations in a number of locations (detailed in specific comments), in particular it would be advantageous for the authors to clearly describe not only the observations made, or what was done, but also what these observations mean or what can be learned from them. One example is the discussion of transport pathways in the abstract (boundary layer and lower FT transport versus upper FT). From the abstract alone it is difficult to understand why these findings are relevant. The authors could adapt these sentences to focus more on how the transport pathways identified in their work helps us better understand the seasonal cycle of Antarctic eBC, for example.

(4) The main point of the paper is not entirely clear from the abstract or conclusions. Several possibilities seem clear, such as the lack of significant trend in eBC concentrations between 2005-2016, or the dominance of biomass burning sources, or the seasonal change in eBC optical properties based on changes in AAE. The paper may benefit from more clear statement of which finding(s) is/are the most novel or interesting outcome of this work. That these findings are presented based on a decade of observations also deserves highlighting. If the authors find it appropriate, they may also consider adapting their title to highlight what was found or learned rather than what was done.

Specific Comments:

P1 L10: Indicate the end date of the analysis (2016?), even if measurements are continuing to present

P1 L17: "internally mixed"

P1 L19: "chemical transport model simulations"

P1 L19-20: What is the implication of direct transport? That higher concentrations arrive at Syowa? That there is less removal? The implications of this transport path may deserve mentioning either here or later in the text

P1 L22: "BC" is used here, but not defined in the abstract

P1 L25: I suggest re-wording this introductory sentence, in particular, "aerosols" cannot be components of "aerosol"

P1 L26: There are many other references and review articles that deserve mentioning here.

P1 L26-27: I appreciate the authors efforts to confront this difficult terminology straight away; however, without acknowledging that soot particles (or carbonaceous particles) are a complex mixture of elemental carbon and organic species this discussion of terminology is somewhat confusing. In addition, defining BC according to only Novakov 1984 misses a large amount of literature on this topic. Finally, using "carbonaceous particles" to describe soot of BC-containing aerosol is confusing because it suggests that non-combustion derived organic species are not types of carbonaceous particles. Several recent reviews provide clear recommendations for consistent terminology in this area.

P2 L3-5: Are there references for this conclusion? Or do the authors find this statement attributable to their own observations? This should be clarified.

P2 L15: This wording is somewhat confusing. Do the authors mean that ice core records of BC are correlated with historical variations in grassland and biofuels BC emissions?

P2 L23-25: The logic here is a bit confusing, presumably anthropogenic BC emissions would have to originate not only from the surface but also from similar geographical regions as biomass burning emissions for this logic to hold. The claim in the previous sentence (i.e., that anthropogenic metals arrive in Antarctica), seems like a stronger

suggestion that anthropogenic BC might be important.

P2 L26: Rather than simply stating that contributions of anthropogenic BC "must be assessed" it would be helpful for the authors to delineate exactly why these sources need to be assessed. This is reasonably well done in the previous sentence.

P2 L38-41: I believe what the authors are trying to say here is that a lot of work has been done on finding the sources of long-range transported mineral particles in Antarctica, but we don't expect BC to have the same sources so we can't apply knowledge from mineral dust to BC. This could be stated much more clearly, potentially at the beginning of the paragraph, if this is the main idea being communicated here.

P2 L41: What is meant by show difficulty?

P2 L42: "Here, we combine.."

P3 L1: "chemical transport model simulations"

P3 L11: 2005 - 2016

P3 L25-26: While a full description of the correction is well placed in the supplement, some estimation of the uncertainty introduced into the measurement from this correction would be useful, along with an estimate of the total uncertainty.

P3 L30: "BC mass"

P3 L34: How frequently was the DL determined? What is the detection limit at a 15 minute time resolution?

P4 L24: Since biomass burning is such a large source of Antarctic BC, was any comparison between biomass burning emission inventories done? Do the authors expect significant differences in their results with different inventories?

P5 L7: What is meant by "frequent"? The mode value?

P5 L10: Showing a plot of the frequency distribution of eBC (e.g., in the supplement) is

more illustrative than reporting the log-normal distribution and fitting parameters in the text.

P5 L13: By "not clear" do the authors mean that there was not statistically significant trend? Can this be shown, for example in the supplement?

P5 L17-19: Is Ferraz at higher altitude or is it just closer to South America? Why is long range transport more likely at Ferraz than Syowa?

P5 L19-20: Are all of these stations using the exact same measurement and definition of BC (or EBC)? Are the same corrections used? This should be discussed, as the same issues have created challenges for comparison of BC measurement in the Arctic (e.g., see Sharma et al., 2017)

P5 21-23: Why might there be more human activity at these two stations? More explanation is useful here.

P5 23-24: What is meant here by "EBC concentrations at the Antarctic coasts in the Indian Ocean sector."? do the authors assert that they are measuring background EBC concentrations not impacted by local human activity? If so, this could be explicitly stated

P5 L29-30: This sentence is somewhat repetitive compared to previous lines.

P5 L30: Use of "this," rather than an explicit statement or a paraphrase of previously presented ideas, makes it difficult to understand what the authors are referring to.

P5 L36-37: This sentence contains problematic wording; it sounds like biomass burning comes from the Antarctic troposphere

P5 L40-42: I believe that the authors are saying that the relative importance of biomass burning will depend on where a particular station is located in Antarctica. This is reasonable but could be made more clear.

P6 L6-7: Mixing month range (here and elsewhere) and season name is a bit confusing, especially since we are discussing the southern hemisphere

P6 L10-20: The discussion of AAE is quite confusing. If an AAE less than one indicates coated eBC, then the authors need to be more clear about what they mean by assess the effects of organic aerosol and mineral particles. Do the authors expect these species to be internally mixed or partially mixed with BC? Or do these species need to be externally mixed to produce an AAE greater than one? This can be explained clearly.

P6 L22-23: It is worth stating here which wavelengths correspond to the UV-IR and Vis-IR channels.

P6 L26-28: This result could use more explanation and support from the literature.

P6 L31: "spring eBC maximum"

P6 L31-32: Since a lot of this discussion depends of knowledge or source regions, the authors may consider moving the discussion of source regions to before the discussion of optical properties and inferences about particle composition

P6 L33: While I agree entirely that the measurements of oxalate are useful to show the seasonal cycle that likely takes place for organic aerosol, it is unclear whether the inference can be made that this indicates more biomass burning organics or BrC. Is there any literature to show that oxalate correlated with brown carbon?

P6 L34: It is somewhat difficult to understand how this conclusion follows from the discussion. The authors need to better explain what it means physically for the AAE_Vis-IR to be negative when the AAE_UV-IR is zero.

P6 L37-38: Was MSA found to be a major aerosol constituent in these filter measurements? If so, these data need to be shown along with the oxalate data.

P7 L1-2: This discussion seems somewhat speculative given that the MSA data is not shown, it is not clear that BrC or any other organic species that could absorb can be

completely discounted here.

P7 L6-8: This statement is based on very little direct evidence (i.e., no direct obser-
vations of MSA or BrC organic aerosol). More evidence needs to be provided, or this
statement needs to be adjusted to better reflect the data presented in this paper.

P7 L15-16: Are 5 day backward trajectories adequate to describe the origins from long
range transport?

P7 L29-30: Doesn't this say that Syowa would be more sensitive to station sources in
winter?

P8 L3-5: Are flow patterns that demonstrate interactions with areas of open ocean
either along the coast or in the MBL associated with times when the authors suggest
that MSA mixing with eBC was important?

P8 L7-13: The authors may consider connecting these observations with those in the
previous section more explicitly.

P8 L15: The authors need to clarify what is meant by "compared" here. How is the
eBC apportioned to these trajectories? Is the dominant trajectory origin assumed to
supply all the eBC in each hour? Or is the eBC apportioned based on the percentage
of each trajectory category for each hour? This should be explained a bit more clearly
in the methods section (and referenced here).

P8 L21: That these stations all use different measurement techniques needs to be dis-
cussed much earlier in the text, when the authors make general comparisons between
BC concentrations measured at different stations.

P8 L29-30: The authors' meaning is not clear in this sentence

P8 L35: "southern Atlantic MBL"

P8 L40: Has MFT been defined?

[Figure]

P9 L1: The wording "difference was similar to differences" is very confusing, and obscures the meaning of this sentence.

P9 L5: Since these trajectories ended in the Southern Ocean, does that suggest the trajectory duration is not sufficient to reach source regions or that marine sources (e.g., ships) could also be important?

P9 L8: What is meant by "seasonal features of eBC concentrations were maximum"?

P9 L10: "in" or "from" the continental FT?

P9 L12: the Antarctic plateau?

P9 L14-16: Transport patterns inferred from CO2 gradients may not be reflective of BC transport pathways, comparing to a shorter lived species may be important to draw these conclusions

P9 L31: "Therefore, we discuss..."

P10 L23-24: I suggest the authors consider re-wording this sentence so that it is not in the form of a question, but rather outlines what was done and why, or the hypothesis.

P10 L33-35: What is learned from these correlations? Would that not suggest that 'other combustion' is more related to anthropogenic activities than biomass burning?

P11 L 5-6: What is being suggested here? Further measurements beyond eBC? This could be explicitly stated.

P11 L9-10: "The following possibilities are contributing factors:..."

P11 L12-13: How does this large precipitation amount impact outflow of BC from source regions? Is part of the seasonal cycle in eBC driven by precipitation removal near the source region or along the transport path? what does the model suggest?

P11 L25: 2005-2016

P11 L30: Are these species internally mixed with eBC?

P11 L31-32: In spring? Or at any time?

P11 L32: "eBC was emitted"

P11 L34: "chemical transport model simulations"

P11 L34-35: These conclusions are somewhat vague. What were the magnitude of these contributions? Is what seasons?

P11 L37-38: This statement should be the other way around: the eBC minimum might be attributable to general transport patterns (higher contributions of the free troposphere and coastal boundary layer)

P12 L6-7: Given wavelengths that these channels correspond to

P18 Figure 1: Labels directly on this map might be more useful than numbers, given that there is ample space and the number of stations is relatively large.

P22 Figure 7: Can a time series of eBC, or AAE be added to the right-hand axis of this plot? Would this shed any light on the influence of marine sources of aerosol absorption characteristics?

---

## Referee Comment (RC2) · Anonymous Referee #2 · 10 Feb 2019

This research paper presents approximately 10 years of black carbon (BC) measurements from Swoya station in coastal eastern Antarctica. It also presents the seasonality in equivalent BC(EBC), its sources during different seasons, comparison of measured data with model and finally determining potential pathways of transport and source regions affecting the station during the observation period. The dataset used in the article is very important in the present scenario due to the importance of BC aerosols/ polar region as well as lesser data availability over the Antarctic region, but the representation is confusing. For easy readability of the article and understanding the conclusions without confusion, I think the restructuring of the different sections of the article is required. I have some general and a few specific comments as detailed below.

General:

[Figure]

1. I would suggest authors provide a clear explanation of data screening procedure (from measured data to useful data for BC). This is very important when we want to find trends and account the data for background concentration values. Authors do refer to an article (Hara et al., 2010) for the data screening procedure but it is not available in that article. Hara et al., 2010 cites Hara et al., 2008 (doi: 10.5194/acpd-8-9883-2008) for these procedures but unfortunately Hara et al., 2008 is not mentioned anywhere in this article, even though it represents the BC concentration from the same location. The best way is to make sure that the data is screened and explained clearly as authors claim it as background values over Antarctica, which in turn would be used by modelers for future studies.

2. I found abstract and conclusion a little confusing, vaguely written, especially with the division of potential source areas (PSA), types of aerosols, and the CHASER model part is not mentioned explicitly. In general, I think abstract and conclusion fails to clearly represent the content of the paper in general, and emphasis on such huge dataset from the remote environment is missing. Acronyms (for ex EBC) appears without expansion, and sentences are misleading (for ex: Fossil fuel combustion in South America and southern Africa also have important contributions).

3. The primary data of the paper is presented in Figure 3. While looking it at it, I got the first impression as high variability in daily EBC values between the years 2005 to 2009, which diminishes after that. It might be possible that more local influences during those years than in other years. I would suggest discussing if any change of location of instrument occurred. Is aethalometer calibrated? Or anything which authors wish to comment on that? Is the sampling line is heated or any changes? I do believe the background values of EBC in the paper, but I think explaining all the above things would make it more satisfactory. 4. Airmass history and classification are explained very nicely, but it stands suddenly out of context. Authors have dedicated Figure 5, Figure 6 and Figure 7 especially for explaining this and they only connect it with EBC in Figure 8. I would suggest the authors make a good connection in meteorology and

aerosol in these figures and sections. Also, classifications and their sub-classifications are not explained and connected for better readability.

5. Based on the CHASER model, EBC origins were classified into three sections (biomass, fossil fuel, and other combustion). Later it is stated that "other combustion" is broadly biomass burning, which makes it as two classifications, which could be inferred from angstrom absorption exponents (AAE) of Figure 3. Is it possible to add information about the mixed state (Internal or external) or aging (fresh or aged) of BC, used in the CHASER model, which showed promising results in monthly values and seasonality of BC in Figure 9?

6. At many places, authors replace "Swoya" with "coastal Antarctica" or "Antarctica coast". It might be useful to replace "Antarctica" in the title with "coastal Antarctica". But I would leave this totally on authors choice. Specific: My suggestion of adding a word(s) is in bold and removing a word(s) is crossed Page 1: Line 10: We measured equivalent black carbon (BC) [ I think you measured BC and corrected it to make it EBC) Line 10: Feb 2005 to Feb 2016 [ adding the end month of measurement] Line 25: the First statement needs a reference Line 34: Antarctic regions; BC concentrations Page 2: The Antarctic is referred to here as one of the remote regions. I think calling it as "Antarctic region" throughout the manuscript is appropriate than "Antarctic regions"? I also think it is worth mentioning and dividing Antarctica as Eastern and Western Antarctica in the introduction as this section talks about tourism and transport from South America and the African region Line 14: please specify where in Antarctica? Page 3: Section 2 heading could be "Measurements, Modelling, and Analysis" Line 5: "Research" is missing in JARE expansion Line 6: It would be worthy to mention the altitude of sampling station at Swoya with latitude and longitude Line 7 To Syowa, the icebreaker ship Shirase approaches every summer (mainly between end-December to – early February) for the transportation of fuel and materials for wintering operations and scientific activity Line 21: I think data screening procedure needs more clarity as it is not in Hara et al., 2010. Hara et al., 2010 cites Hara et al., 2008 and I would suggest

citing the right paper here. Line 23 to 34: what is the value of multiple scattering and loading parameter used in making BC to EBC? Would be a good idea to mention it explicitly here. Line 28: Attenuation at 880 nm is used widely for BC retrievals, I would suggest making changes in the statement accordingly Line 35: the statement "We use a multi-wavelength……." could be rephrased like "using the spectral (or multi-wavelength) aerosol absorption values retrieved from aethalometer, we estimated AAE Page 4: Line 10-14: CHASER could be expanded. There are other acronyms also need to be expanded. I think it would be helpful for readers who are not modelers. Line 39: "cooking" not "cocking" Page 5: Section heading "Discussions" Line13-14: Any seasonal long-term trend at Swoya? It would be worth seeing whether there is an increase or decrease in summer (like Neumayer) or spring Line 25-42: It is not much clear. I would suggest defining seasons and maintain uniformity in the discussion of seasonality and comparison with other stations. The possible sources in each season could be also be highlighted. Page 6: Line 21: BrC Brown Carbon(BrC) Line 29: larger negative values (<-0.4) Line 24: What is a high correlation means here? R2 values are lesser for June-Aug, in comparison to other months Page 7: Line 6: Slopes >1 but AAE was lesser in spring (Figure 3d), so how you suggest it is biomass burning aerosol of organic origin? Please clarify Line 15: "First, we compare EBC data to the air mass history at Syowa" this line does not seem appropriate here

Line 17: for the 3rd classification, do you mean outflow from the high-latitude Antarctic continent to coastal Antarctica? Please clarify

Line30: It appears that the probability density of air mass arriving at Swoya shows an East-West spread from one month to another month, as compared to North-South spread. In that case, transport from inland Antarctica is more important than long-range transport from populated continents. Is it the case?

Line 41: could specify an approximate tropopause height.

Line 35-43: From Figure 6, it appears that Swoya is influenced by high-latitude inland

Antarctica airmass during all months (which is relatively less in January). What is the final take from Figure 6?

Page 8:

Line 3-5: Is this subclassification of the classification on page 7(Line 16-18)? I think it is 2 sub-classifications of the previous classification, but it is not clear in the text.

Line 6: classification of airmass origin > 75S could be renamed as remote continental or Antarctic continental, as naming it continental confuses with polluted and populated continents.

Line 8: statement is not clear

Line 29-30: Is an increase in MBL airmass origin EBC, could be due to Ship emissions in the Antarctic circle (for fishing or tourism)?

Page 9:

Line 25-26: Filter biased problems and related uncertainty were not discussed while detailing EBC. I am glad that the authors bring it up here.

Line 28-29: This is already discussed in section 3.1

Line 34-36: So basically, this classification is biomass and fossil fuel? As authors said other combustion is also biomass in the broad sense, so how this is different than AAE of section 3.1, besides it is from CHASER model?

Line 37-39: As the text says biomass burning is dominant in spring, and Figure 8 says it is Marine BL and Marine FT contributing to the EBC at Swoya, so what would be the conclusion? It is not clear

Page 10:

Line 1: It is difficult to identify the August-October peak in Figure 10b

Line 6: "contribution of BB" or "contribution from BB". Also, the difference is significant if

you compare magnitudes of BC, which showed a 25 to 50 % decrease from 2011-2012 to 2015-2016

Line 24-25: I am not clear about the statement which ends with a question mark. The explanation is given in the next lines and I consider that statement as a misfit.

Is it possible to use a symbol for "BB-model-BC concentrations", something like the BB(BC) model? Similarly, for FFC and OC

Line 30-35: It is quite difficult to follow the month to month explanation from Figure 10. Authors should either include minor labels or ticks or any other way to the identification

The authors might consider a stacked column chart (by normalizing it with total concentrations) for all panel 10b, c, and d. So, the stacked column length would be total, partitions in the column would represent the contributions of South America, Southern Africa, and Australia. I think in that way, all the description in the text would be clearer. But I leave this to authors.

Page11:

Line 1-6; I think as the southern America coastline extends much to the southern latitudes (near to western Antarctic peninsula), and the westward transport along coastal Antarctica, might be also a reason for the higher influence on the Antarctic BC, in addition to the GDP. This could be also clarified and detailed in the manuscript.

Conclusion section: It should be rephrased to highlight important data set period, seasonality of BC, transport patterns at Swoya, model comparison and regional contribution from South America, South Africa, and Australia.

Figure 1: I think it would be better to place Swoya station as a different symbol or by placing the name next to the current symbol. Identifying regions like South America and Africa could be also a good idea as it comes quite often in the manuscript Figure 2: I would suggest adding first and last labels in the Y axis too. Authors may consider writing Swoya near the red circle. Figure 3: Y-axis scale for panel c is missing. It

Interactive
comment

doesn't seem matching with panel a. Blue line mentioned in the caption is not visible in the panel a Figure 5: Swoya location could be shown in a different color/symbol for better visibility. I would suggest using a latitude scale too for this figure Figure 9: In panel b, the regression coefficient could be shown Figure 10: Caption for the panels are not clear. Do authors mean BB aerosols from South America as a whole or BB from South America, contributing to EBC at Swoya.

List of Acronyms: Some Acronyms from the manuscript are missing in the list, like HYSPLIT, CHASER.

---

## Author Comment (AC1) · 18 Mar 2019

Reply to comments from Referee #1: We would like to thank your helpful and complaisant comments to improve our manuscript. All comments are responded and addressed in the revised manuscript. Details are listed as follows. Corrected parts on comments from Ref. #1 were marked by red characters in the revised manuscript.

Comment from Referee: In particular there is a tendency toward paragraphs that focus on multiple ideas and that lack clear topic sentences or conclusions. An examination of the order in which ideas are presented may also be helpful.

Reply from Authors: We rewrote abstract and conclusions to be clear. Because one of reasons for lack clear topic sentences may be length of paragraph, we divided long

paragraph into a few paragraphs. Additionally, some results and discussion (e.g., AAE) moved to last section.

Comment from Referee: I suggest that the Authors ensure that their stated conclusions follow directly from the data they have presented.

Reply from Authors: We rewrote conclusions on basis of results from field measurements and model simulation.

Comment from Referee: In particular it would be advantageous for the authors to clearly describe not only the observations made, or what was done, but also what these observations mean or what can be learned from them.

Reply from Authors: We added and modified text on basis of your specific comments.

Comment from Referee: The main point of the paper is not entirely clear from the abstract or conclusions.

Reply from Authors: We arranged and rewrote sentences in abstract and conclusions on basis of comments from you and Referee #2.

Comment from Referee: P1 L10: Indicate the end date of the analysis (2016?), even if measurements are continuing to present.

Reply from Authors: BC measurements have been made at Syowa station. In this study, we used BC data in 2005-2016. This was added in abstract.

Comment from Referee: P1 L17: "internally mixed"

Reply from Authors: Because of modification of abstract, this words was removed.

Comment from Referee: P1 L19: "chemical transport model simulations"

Reply from Authors: We changed to "chemical transport model simulations" here and in the other locations.

Comment from Referee: P1 L19-20: What is the implication of direct transport? That

higher concentrations arrive at Syowa? That there is less removal? The implications of this transport path may deserve mentioning either here or later in the text

Reply from Authors: To avoid confusion, we changed "direct transport" to just "transport" or more specific explanations.

Comment from Referee: P1 L22: "BC" is used here, but not defined in the abstract

Reply from Authors: BC was defined in abstract of the revised manuscript, although location of BC in the text was changed by modification of manuscript.

Comment from Referee: P1 L25: I suggest re-wording this introductory sentence, in particular, "aerosols" cannot be components of "aerosol"

Reply from Authors: We changed into "Carbonaceous aerosols are major aerosols in the troposphere".

Comment from Referee: P1 L26: There are many other references and review articles that deserve mentioning here.

Reply from Authors: We added some references such as Andreae and Gelencsér, 2006; Bond et al., 2013.

Comment from Referee: P1 L26-27: Without acknowledging that soot particles (or carbonaceous particles) are a complex mixture of elemental carbon and organic species this discussion of terminology is somewhat confusing. In addition, defining BC according to only Novakov misses a large amount of literature on this topic. Several recent reviews provide clear recommendations for consistent terminology in this area.

Reply from Authors: In addition to definition by Novakov (1984), we added some references (e.g., Andreae and Gelencsér, 2006; Bond et al., 2013). Short explanation of definition about carbonaceous aerosols and BC were added into the text, as follows.

Apart from secondary organics associated with biogenic cycles, most of carbonaceous aerosols (e.g., soot) can be released from combustion of biomass and fuels. Soot

particles consist of refractory and insoluble matter (aka EC) and organics (e.g., An-dreae and Gelencsér, 2006). As defined by Novakov (1984), BC comprises particulate graphitic particles. Recently, BC is defined by the following physical properties; (1) strong light absorption, (2) refractory, (3) insoluble, (4) aggregates of small carbon spherules (e.g., Bond et al., 2013).

Comment from Referee: P2 L3-5: Are there references for this conclusion? Or do the Authors find this statement attributable to their own observations? This should be clarified.

Reply from Authors: This is consideration by researchers (including us) who observed BC in the Antarctica. We changed this sentence to " It has been considered that BC must be supplied from outside of Antarctica, i.e. long-range transport, to maintain the background BC level and that it has seasonal features in the Antarctic atmosphere because of the low BC source strength in the Antarctic region. "

Comment from Referee: P2 L15: This wording is somewhat confusing. Do the Authors mean that ice core records of BC are correlated with historical variations in grassland and biofuels BC emissions?

Reply from Authors: To avoid confusion, this sentence changed to " In fact, BC records for the past 150 years in the Antarctic ice cores (WAIS core in Western Antarctica and Low dome core in Eastern Antarctica) showed influences by El Niño-Southern Oscillation (ENSO) and BC emissions from biomass burning and human activity in the source areas (Bisiaux et al., 2012)."

Comment from Referee: P2 L23-25: The logic here is a bit confusing, presumably anthropogenic BC emissions would have to originate not only from the surface but also from similar geographical regions as biomass burning emissions for this logic to hold. The claim in the previous sentence (i.e., that anthropogenic metals arrive in Antarctica), seems like a stronger suggestion that anthropogenic BC might be important.

[Figure]

Reply from Authors: We changed the explanation to " In contrast to the Arctic atmosphere, earlier works concluded that anthropogenic effects were only slight and negligible for aerosols in the Antarctic atmosphere (e.g., Weller et al., 2011, 2013), although some anthropogenic metals such as Pb have been found in snow and ice cores in the Antarctic region (Planchon et al., 2002; Vallelonga et al., 2002). Considering that biomass burning occurs on the ground in forests and grasslands, anthropogenic BC (derived mainly from fossil fuel combustion) can be transported to Antarctica."

Comment from Referee: P2 L26: Rather than simply stating that contributions of anthropogenic BC "must be assessed" it would be helpful for the Authors to delineate exactly why these sources need to be assessed. This is reasonably well done in the previous sentence.

Reply from Authors: We changed to "must be assessed", here.

Comment from Referee: P2 L38-41: We don't expect BC to have the same sources so we can't apply knowledge from mineral dust to BC. This could be stated much more clearly, potentially at the beginning of the paragraph, if this is the main idea being communicated here.

Reply from Authors: We changed the explanation in the paragraph as follows.

To elucidate BC transport from the low latitudes and mid-latitudes to the Antarctic region, we must ascertain the potential source area (PSA) and transport pathway. Actually, BC cannot be vaporized in ambient conditions. Therefore, BC must be transported from the origins (i.e. combustion processes) to the Antarctica. However, chemical analyses such as isotope ratio investigations are difficult to apply for identification of BC origins because the major BC component is graphite. Hara et al. (2010) described BC transport from South America and southern Africa to Syowa Station, Antarctica. Similarly to BC, mineral particles are transported from their origins to Antarctica, except for local emissions originating within the Antarctic Circle. For identification of the origins of mineral particles, earlier studies have been conducted to analyze and assess

[Figure]

PSA of mineral particles based on Nd/Sr isotope ratios (Smith et al., 2003; Delmonte et al., 2004, 2008; Bory et al., 2010; Valleloga et al., 2010; Aarons et al., 2016), Pb isotope ratios (De Deckker et al., 2010; Gilli et al., 2016), rare earth element patterns (Gabrielli et al., 2010; Valleloga et al., 2010; Wegner et al., 2012, Aarons et al., 2016), and trajectory/models (Perreira et al., 2004; Li et al., 2008; Albani et al., 2010; Gasso et al., 2010; Krinner et al., 2010; Neff and Bertler, 2015). From the aspect of mineral particles transported into Antarctica, South America (mostly Patagonia) has been identified as the most dominant PSA, whereas Australia and Africa respectively show minor and unimportant PSAs (e.g., Neff and Bertler, 2015). Although one must consider the following differences between BC and minerals, (1) geographical locations of PSA, (2) seasonality of source strength, and (3) size of aerosol particles containing BC and minerals, BC can be transported by outflow from the continents in the mid-latitudes to Antarctica. Here, we combine BC measurements with backward trajectory and chemical transport model simulation. This study was conducted to elucidate BC origins and PSA and to characterize BC concentrations and their seasonal features at Syowa Station, Antarctica located in the Indian Ocean sector.

Comment from Referee: P2 L41: What is meant by show difficulty?

Reply from Authors: Because of modification of text, this was removed from the text.

Comment from Referee: P2 L42: "Here, we combine..."

Reply from Authors: This was changed based on your suggestion.

Comment from Referee: P3 L1: "chemical transport model simulations"

Reply from Authors: This was changed based on your suggestion.

Comment from Referee: P3 L11: 2005 – 2016

Reply from Authors: This was changed based on your suggestion.

Comment from Referee: P3 L25-26: While a full description of the correction is well

placed in the supplement, some estimation of the uncertainty introduced into the measurement from this correction would be useful, along with an estimate of the total uncertainty.

Reply from Authors: The following explanation was added into the end of this paragraph.

Uncertainty of the measured EBC concentrations relates to (1) stability of the optical signal, (2) flow rate control, (3) spot area, and (4) scattering and shadowing effects. The detection limit value corresponds to uncertainty resulting from processes of (1)–(3). Uncertainty by the process (4) depends on the aerosol number concentrations and optical properties (single scattering albedo). The EBC concentrations corrected using Weingartner's method were mostly lower by 0.5–2% compared to the uncorrected EBC concentrations in this study (Fig. S1). Less difference between the corrected and uncorrected EBC concentrations might derive from higher single scattering albedo and replacement of the filter spot before optical attenuation reaching to 10% in most cases in our measurement conditions at Syowa.

Comment from Referee: P3 L30: "BC mass"

Reply from Authors: This was changed based on your suggestion.

Comment from Referee: P3 L34: How frequently was the DL determined? What is the detection limit at a minute time resolution?

Reply from Authors: DL was determined several times during our measurements. In this study, we did not DL at one-minute resolution, because EBC concentrations were too low to be measured with time resolution of one minute unlike urban areas.

Comment from Referee: P4 L24: Since biomass burning is such a large source of Antarctic BC, was any comparison between biomass burning emission inventories done? Do the Authors expect significant differences in their results with different inventories?

[Figure]

Reply from Authors: Regional biomass burning emission and its seasonal trend in MACC are similar to those in other inventory (e.g., GFED), although there are slight difference in regional distribution of biomass burning in each inventory. We added this into the text in the revised manuscript.

Comment from Referee: P5 L7: What is meant by "frequent"? The mode value?

Reply from Authors: Modal value is correct. We change from "frequent" to "Modal".

Comment from Referee: P5 L10: Showing a plot of the frequency distribution of eBC (e.g., in the supplement) is more illustrative than reporting the log-normal distribution and fitting parameters in the text.

Reply from Authors: The histogram of EBC concentrations with regression line was added in Figure of Supplementary (Fig. S2).

Comment from Referee: P5 L13: By "not clear" do the Authors mean that there was not statistically significant trend? Can this be shown, for example in the supplement?

Reply from Authors: We added results and figure of trend analysis in supplementary. On basis of the results, we added the following explanation in the revised manuscript.

From trend analysis (Supplementary and Fig. S3), a very slight decreasing trend (-0.036 ng m-3 yr-1, p = 0.0145) was observed in our measurements for 2005–2016. However, an increasing trend (0.105 ng m-3 yr-1, P < 0.001) was obtained in 2010–2016. These trend values included temporal trends, as explained below. Therefore, we concluded only slightly whether these trends were long-term EBC trends or not. More continuous EBC measurements must be taken at Syowa Station to analyze long-term trends.

Comment from Referee: P5 L17-19: Is Ferraz at higher altitude or is it just closer to South America? Why is long range transport more likely at Ferraz than Syowa?

Reply from Authors: Ferraz is located at northern part of the Antarctic Peninsula closer

to the South America. In addition, air masses at Ferraz were transported frequently from South America (Pereira et al., 2004), so that the long-range transport from South America might engender higher EBC concentrations at Ferraz than those in the other Antarctic coasts. This explanation was added in the text.

Comment from Referee: P5 L19-20: Are all of these stations using the exact same measurement and definition of BC (or EBC)? Are the same corrections used? This should be discussed, as the same issues have created challenges for comparison of BC measurement in the Arctic (e.g., see Sharma et al., 2017).

Reply from Authors: We corrected EBC concentrations in this study. However, EBC (or BC) concentrations measured at other Antarctic stations were not corrected. Also, difference between the corrected and uncorrected EBC concentrations at Syowa Station was discussed in the revise manuscript, as follows.

In earlier works, EBC concentrations were uncorrected, unlike this study. The EBC concentrations corrected using Weingartner's method decreased mostly by 0.5–2% compared to the uncorrected EBC concentrations in this study (Fig. S2). The lesser difference between the corrected and uncorrected EBC concentrations might result from (1) higher single scattering albedo and (2) replacement of filter spot before optical attenuation reaching to 10% in most cases in our measurement conditions at Syowa. Therefore, we can compare EBC concentrations in this study to the uncorrected EBC concentrations measured at other Antarctic stations in previous works.

Furthermore, some discussion about comparison of BC measurement in the Arctic (e.g., see Sharma et al., 2017) was added in the text, as follows.

In addition to filter-based EBC measurements, a single particle soot photometer (SP2) has been used for the measurement of refractory BC (rBC) (e.g., Bond et al., 2013; Sharma et al., 2017). According to Sharma et al. (2017), high correlation with R2 = 0.8–0.9 and slopes = 1.2–1.6 was observed between rBC and EBC in aerosols in the Arctic, where aerosol concentrations and anthropogenic effects were greater and

stronger than those in Antarctica. Considering different conditions of aerosol chemistry and optical properties between those in Antarctica and the Arctic, correlation in the Antarctica is expected to be different from that in the Arctic. No report of the relevant literature has described SP2 used to measure rBC year-round in the Antarctic region. Because of higher single-scattering albedo and lower aerosol concentrations in the Antarctica, differences between rBC and EBC might not be greater than in the Arctic.

Comment from Referee: P5 21-23: Why might there be more human activity at these two stations? More explanation is useful here.

Reply from Authors: High BC concentrations at Maitri and Larsemann Hills did not result from not more human activity but insufficient screening of local contaminated data. This was added in the text.

Comment from Referee: P5 23-24: What is meant here by "EBC concentrations at the Antarctic coasts in the Indian Ocean sector."? do the Authors assert that they are measuring background EBC concentrations not impacted by local human activity? If so, this could be explicitly stated

Reply from Authors: We mean "EBC concentrations observed at Syowa corresponded to background EBC concentrations at the Antarctic coasts in the Indian Ocean sector." The sentence was modified.

Comment from Referee: P5 L29-30: This sentence is somewhat repetitive compared to previous lines.

Reply from Authors: This repetition was removed from the text.

Comment from Referee: P5 L30: Use of "this," rather than an explicit statement or a paraphrase of previously presented ideas, makes it difficult to understand what the authors are referring to. Reply from Authors: Instead of "this", we used specific statement in the revised manuscript.

Comment from Referee: P5 L36-37: This sentence contains problematic wording; it

sounds like biomass burning comes from the Antarctic troposphere

Reply from Authors: This sentence changed to "biomass burning in the mid- and low latitudes has been regarded as having dominant origins of EBC measured in the Antarctic troposphere".

Comment from Referee: P5 L40-42: I believe that the Authors are saying that the relative importance of biomass burning will depend on where a particular station is located in Antarctica. This is reasonable but could be made more clear.

Reply from Authors: This sentence changed to "the contribution of biomass burning from each PSA likely depends on where the respective coastal stations are located (e.g., sectors of Atlantic, Indian, and Pacific Oceans)."

Comment from Referee: P6 L6-7: Mixing month range (here and elsewhere) and season name is a bit confusing, especially since we are discussing the southern hemisphere

Reply from Authors: We showed month and season in the revised manuscript.

Comment from Referee: P6 L10-20: The discussion of AAE is quite confusing. If an AAE less than one indicates coated eBC, then the Authors need to be more clear about what they mean by assess the effects of organic aerosol and mineral particles. Do the Authors expect these species to be internally mixed or partially mixed with BC? Or do these species need to be externally mixed to produce an AAE greater than one? This can be explained clearly.

Reply from Authors: On basis of comments from you and referee #2, we corrected the discussion and statement as follows.

The CHASER model also indicates that internal mixing states of BC were dominated through the year (not shown, details published elsewhere). Therefore, lower AAE in April–October might result from the dominant presence of coated EBC particles (internal mixtures) at Syowa. The slight AAE increase corresponded to the spring maximum

of EBC concentrations. The following possibilities were considered for the slight AAE increase: (1) change of mixing states of BC and (2) contribution of other light-absorbing materials such as organic aerosols and minerals. The organic aerosols and minerals have high AAE, for instance, 3.5–7 for organics and typically 2–3 for minerals (e.g., Bond et al., 2013, and references therein). Although the internal mixing states of BC were dominant in CHASER model simulation, external mixtures of BC increased in spring EBC maximum (not shown, details published elsewhere). Considering AAE of external mixing of BC, increase of external mixing of BC can engender an AAE increase. Additionally, spring EBC maximum at the Antarctic coasts was associated closely with biomass burning. Organic aerosols with high AAE derived from biomass burning were expected to be transported simultaneously into Antarctica. Consequently, transport of organic aerosols might contribute to the slight AAE increase in September – October.

Comment from Referee: P6 L22-23: It is worth stating here which wavelengths correspond to the UV-IR and Vis-IR channels.

Reply from Authors: Wavelengths were added into the text.

Comment from Referee: P6 L26-28: This result could use more explanation and support from the literature.

Reply from Authors: We changed the explanation as follows.

High optical absorption by organic aerosols in the UV ranges engenders an increase of AAEUV-IR and the higher ratios (e.g., Bond et al., 2013). Therefore, the difference suggests that organic aerosols, rather than effects of mineral particles, contributed to optical absorption and AAE.

Comment from Referee: P6 L31: "spring eBC maximum"

Reply from Authors: This was changed based on your suggestion.

Comment from Referee: P6 L31-32: Since a lot of this discussion depends of knowledge or source regions, the Authors may consider moving the discussion of source regions to before the discussion of optical properties and inferences about particle composition

Reply from Authors: Paragraph on AAE and optical properties were moved to last section.

Comment from Referee: P6 L33: While I agree entirely that the measurements of oxalate are useful to show the seasonal cycle that likely takes place for organic aerosol, it is unclear whether the inference can be made that this indicates more biomass burning organics or BrC. Is there any literature to show that oxalate correlated with brown carbon?

Reply from Authors: Yes. Previous work (Zhang et al. 2012) showed that high concentrations of oxalate and brown carbons were associated with secondary organic aerosol formation in a condensed phase. This work was referred in the revised manuscript. Also, short explanation was added to discussion.

Comment from Referee: P6 L34: It is somewhat difficult to understand how this conclusion follows from the discussion. The Authors need to better explain what it means physically for the AAE_Vis- IR to be negative when the AAE_UV-IR is zero.

Reply from Authors: We changed the explanation as follows.

Additionally, high concentrations of oxalate and brown carbons are associated with secondary organic aerosol formation in a condensed phase (e.g., Zhang et al., 2012). When optical absorption in UV regions was increased by organic aerosols (i.e. BrC), correlation between AAEUV-IR and AAEVis-IR can be shifted to larger AAEUV-IR region. This change might engender larger negative intercept values. Therefore, the larger negative intercepts in October–November might result from effects of organic aerosols derived from biomass burning.

Comment from Referee: P6 L37-38: Was MSA found to be a major aerosol constituent

in these filter measurements? If so, these data need to be shown along with the oxalate data.

Reply from Authors: MSA feature was added to Fig. S6 which showed the oxalate data.

Comment from Referee: P7 L1-2: This discussion seems somewhat speculative given that the MSA data is not shown, it is not clear that BrC or any other organic species that could absorb can be completely discounted here.

Reply from Authors: We modified the sentence into "Therefore, AAE in the summer (December–February) might be associated with EBC aging processes and with the presence and mixing of organic aerosols (e.g. CH3SO3H) derived from oceanic bioactivity."

Comment from Referee: P7 L6-8: This statement is based on very little direct evidence (i.e., no direct observations of MSA or BrC organic aerosol). More evidence needs to be provided, or this statement needs to be adjusted to better reflect the data presented in this paper.

Reply from Authors: We added direct evidence of MSA feature in Supplementary. The explanation was changed into "The concentrations of EBC and organic aerosols derived from biomass burning increased in the spring maximum as described above, whereas the EBC concentrations decreased and the concentrations of organic aerosols such as CH3SO3- derived from oceanic bioactivity increased during summer."

Comment from Referee: P7 L15-16: Are 5 day backward trajectories adequate to describe the origins from long range transport?

Reply from Authors: From aspect of identification of PSA, 5 days are not enough. Because uncertainty of trajectory analysis particularly in troposphere tends to be larger in longer analysis time. In this study, we calculated 5 day backward trajectory. However, the trajectory can provide important information on transport pathway, for example via

MBL and continental FT.

Comment from Referee: P7 L29-30: Doesn't this say that Syowa would be more sensitive to station sources in winter?

Reply from Authors: We did not think so. Although density of air masses from the Antarctic continent increased in winter, transport from ocean (mid-latitude) was major transport pathway.

Comment from Referee: P8 L3-5: Are flow patterns that demonstrate interactions with areas of open ocean either along the coast or in the MBL associated with times when the Authors suggest that MSA mixing with eBC was important?

Reply from Authors: In summer, transport from open water area (mid-latitude) showed high density as shown in Fig. 4. This air mass history might be associated with MSA concentrations at Syowa and mixing states of aerosol particles containing EBC. However, this explanation was not added in the text because discussion on AAE was move to last section, and lengthy explanation should be avoided.

Comment from Referee: P8 L7-13: The Authors may consider connecting these observations with those in the previous section more explicitly.

Reply from Authors: To explain more explicitly, we changed these sentence as follows.

As described above, EBC is expected to be supplied mostly from outside of Antarctica. Plausible transport pathways are transport via MBL and FT. We must know the EBC concentrations of each air mass origin (MBL, coastal BL, continental BL, continental FT, coastal FT, and MFT) to elucidate EBC transport pathway to the Antarctica. Figure 6 depicts seasonal features of air mass origins in each month and monthly mean and median EBC concentrations at Syowa during our measurements (2005–2016). The dominant air mass origins were MBL, coastal BL, coastal FT, and continental FT. The most dominant air mass origins were MBL and coastal BL in November–February. In addition to MBL and coastal BL, the contributions of transport from coastal FT and

continental FT increased in February/March – October at Syowa, although year-to-year differences were found in the seasonal variations of air mass origins. Particularly, the contribution of transport from continental FT in March–October was higher than that in other years. This change corresponded to lower EBC concentrations in July of 2010–2012, as described above. Therefore, the increasing trend of EBC concentrations in July of 2010–2016 might not be a long-term trend but a temporal trend resulting from year-to-year variations of air mass history.

Comment from Referee: P8 L15: The Authors need to clarify what is meant by "compared" here. How is the eBC apportioned to these trajectories? Is the dominant trajectory origin assumed to supply all the eBC in each hour? Or is the eBC apportioned based on the percentage of each trajectory category for each hour? This should be explained a bit more clearly in the methods section (and referenced here).

Reply from Authors: The sentences were modified as follows.

For comparison between EBC concentrations and air mass origins, hourly mean EBC concentrations were estimated. Then, hourly EBC data were classified into each air mass origin.

Comment from Referee: P8 L21: That these stations all use different measurement techniques needs to be discussed much earlier in the text, when the Authors make general comparisons between BC concentrations measured at different stations.

Reply from Authors: This explanation was moved to discussion on Fig. 3 (seasonal features of EBC conc.). Some explanation about EBC data at other stations were also added in the text.

Comment from Referee: P8 L29-30: The Author's meaning is not clear in this sentence

Reply from Authors: This sentence was changed as follows.

Seasonal features of EBC concentrations in MBL (Fig. 7a) might correspond to those in the MBL in the Southern Ocean in Atlantic and Indian sectors, considering that air

masses were transported dominantly via MBL from the mid-latitudes by the cyclone approach.

Comment from Referee: P8 L35: "southern Atlantic MBL"

Reply from Authors: This was changed based on your suggestion.

Comment from Referee: P8 L40: Has MFT been defined?

Reply from Authors: Not yet. I defined MFT in the revised manuscript.

Comment from Referee: P9 L1: The wording "difference was similar to differences" is very confusing, and obscures the meaning of this sentence.

Reply from Authors: To avoid confusion, this sentence was changed as follows.

Seasonal variations of CO concentrations and fire counts (Gros et al., 1999; Edwards et al., 2006a, 2006b) were similar to the seasonal features of EBC concentrations at Syowa, as described above.

Comment from Referee: P9 L5: Since these trajectories ended in the Southern Ocean, does that suggest the trajectory duration is not sufficient to reach source regions or that marine sources (e.g., ships) could also be important?

Reply from Authors: Ship operation in MBL can release EBC to the atmosphere. Density of marine traffic (i.e. ship operation) in the Southern Ocean and near the Antarctic coasts was too low to engender increase of EBC concentrations in air mass from MBL, although ship emission can have an influence locally on EBC concentrations, for examples ship-borne tourism in the Antarctic Peninsula during summer. These explanations were added into the revised manuscript.

Comment from Referee: P9 L8: What is meant by "seasonal features of eBC concentrations were maximum"?

Reply from Authors: "seasonal features of EBC concentrations had maximum in

October–" is correct. This was modified in the revised manuscript.

Comment from Referee: P9 L10: "in" or "from" the continental FT?

Reply from Authors: "from" is correct. This was modified in the revised manuscript.

Comment from Referee: P9 L12: the Antarctic plateau?

Reply from Authors: This was changed based on your suggestion.

Comment from Referee: P9 L14-16: Transport patterns inferred from $CO_2$ gradients may not be reflective of BC transport pathways, comparing to a shorter lived species may be important to draw these conclusions

Reply from Authors: This explanation was removed from the manuscript.

Comment from Referee: P9 L31: "Therefore, we discuss. . ."

Reply from Authors: This was changed based on your suggestion.

Comment from Referee: P10 L23-24: I suggest the Authors consider re-wording this sentence so that it is not in the form of a question, but rather outlines what was done and why, or the hypothesis.

Reply from Authors: This sentence was changed as follows.

We must ascertain the transport pathway from Australia to the Syowa to understand the high BB-mBC concentrations in Australia.

Comment from Referee: P10 L33-35: What is learned from these correlations? Would that not suggest that 'other combustion' is more related to anthropogenic activities than biomass burning?

Reply from Authors: The following sentence was added here.

The good correlation found between FFC-mBC and OC-mBC implies strongly that seasonal features of FFC-mBC and OC-mBC might reflect variations of transport strength

from each PSA to Syowa.

Comment from Referee: P11 L 5-6: What is being suggested here? Further measurements beyond eBC? This could be explicitly stated.

Reply from Authors: We modified the sentence as follows.

Therefore, continual EBC measurements must be conducted at the Antarctic coasts to monitor the atmospheric substances (e.g. EBC) originating from combustion in the Southern Hemisphere.

Comment from Referee: P11 L9-10: "The following possibilities are contributing factors:. . ."

Reply from Authors: This was changed based on your suggestion.

Comment from Referee: P11 L12-13: How does this large precipitation amount impact outflow of BC from source regions? Is part of the seasonal cycle in eBC driven by precipitation removal near the source region or along the transport path? what does the model suggest?

Reply from Authors: We added the following explanation.

In the CHASER model, fresh BC immediately after release from BB is assumed as hydrophobic, so that few fresh BC might be removed by precipitation near source areas. Aging processes during transport can engender gradual change into internal mixtures of BC with a hydrophilic surface. Then, BC can be scavenged through wet deposition during transport. Considering clear seasonal features of CO with longer residence time than BC (Edwards et al., 2006a, 2006b; van den Werf et al., 2006), seasonal variation of BC emissions might have a greater contribution to seasonal features of mBC and EBC at Syowa than wet deposition of BC during transport.

Comment from Referee: P11 L25: 2005-2016

Reply from Authors: This was changed based on your suggestion.

Comment from Referee: P11 L30: Are these species internally mixed with eBC?

Reply from Authors: Aethalometer cannot provide ambient mixing states of aerosols. Thus, we need to compare to earlier work (e.g., Ueda et al., 2018). However, this explanation was removed from the text in conclusions because of modification of the sentences in conclusions.

Comment from Referee: P11 L31-32: In spring? Or at any time?

Reply from Authors: We mean winter – spring. This was added in the text of conclusion.

Comment from Referee: P11 L32: "eBC was emitted"

Reply from Authors: This was changed based on your suggestion.

Comment from Referee: P11 L34: "chemical transport model simulations"

Reply from Authors: This was changed based on your suggestion.

Comment from Referee: P11 L34-35: These conclusions are somewhat vague. What were the magnitude of these contributions? Is what seasons?

Reply from Authors: To avoid length explanation, we changed as follows.

The CHASER model simulation showed that the most important origins and PSA of EBC at Syowa Station were biomass burning in South America and southern Africa.

Comment from Referee: P11 L37-38: This statement should be the other way around: the eBC minimum might be attributable to general transport patterns (higher contributions of the free troposphere and coastal boundary layer)

Reply from Authors: This was changed based on your suggestion.

Comment from Referee: P12 L6-7: Given wavelengths that these channels correspond to

Reply from Authors: Wavelengths were added in the revised manuscript.

Comment from Referee: P18 Figure 1: Labels directly on this map might be more useful than numbers, given that there is ample space and the number of stations is relatively large.

Reply from Authors: Labels of each station were added in Fig. 1.

Comment from Referee: P22 Figure 7: Can a time series of eBC, or AAE be added to the right-hand axis of this plot? Would this shed any light on the influence of marine sources of aerosol absorption characteristics?

Reply from Authors: We added seasonal features of EBC concentrations in this figure (Fig. 6 in the revised manuscript).

Please also note the supplement to this comment:
https://www.atmos-chem-phys-discuss.net/acp-2018-1190/acp-2018-1190-AC1-supplement.pdf

———————————————————

[Figure]

**Supplement:**

[revised manuscript text omitted]

$$R = \frac{1}{\left(\frac{1}{f}-1\right)\frac{lnATN-\ln(10\%)}{ln(50\%)-ln(10\%)}+1} \tag{4}$$

Correction parameter $f$ is estimated using the following relation.

$$f = a(1-\omega_0)+1 \tag{5}$$

In that equation, $\omega_0$ is the single-scattering albedo. The determined parameters *a* are 0.87 ($\lambda = 450$ nm) and 0.85 ($\lambda = 660$ nm). Because of the low BC concentrations detected at Antarctic coasts, $\omega_0$ was found mostly as 0.97–0.99 at Syowa Station (Yabuki et al., preparation for publication). Similar values of $\omega_0$ were measured also at Neumayer (Weller et al., 2013). Therefore, *f* values can be 1.02–1 using our measurement conditions. Here, we used 1.01 as the *f* values. Furthermore, $\Delta t$ was 120 min (2 hr) in this study because of lower BC concentrations. The corrected BC concentrations were estimated for all BC data (recorded every 15 min) in this study. In aethalometer measurements using AE31, $\sigma_{ATN}$ is 14,625 nm m$^2$ g$^{-1}$ $\lambda^{-1}$.

[Figure]

**Figure S1: Histogram of correction factor of Weingartner's correction in aethalometer data measured at Syowa Station, Antarctica during our measurements.**

[Figure]

**Figure S2: Histogram of EBC concentration measured at Syowa Station, Antarctica in 2005–2016.**

[Figure]

**Figure S3: Variations of monthly median of EBC anomaly at Syowa Station, Antarctica. Red, blue, and green lines respectively present regression lines in all periods (2005–2016) and 2010–2016, and anomaly value of 1.**

In our EBC dataset, long gaps in data exist because of instrumental troubles in 2007 and Jan. – March in 2011; short data gaps were caused by local contamination. In addition, EBC concentrations showed strong seasonal variations as depicted in Fig. 3. First, one must remove the strong seasonality before analysis of long-term trends. Daily median EBC concentrations during our measurements were estimated as normal-like values of EBC concentrations at Syowa Station. Then, the ratios of ambient EBC concentrations to the normal-like EBC concentrations were calculated and were used as anomaly values. The monthly median EBC anomaly is shown in Fig. S3. Linear regression lines were fitted during all periods (2005–2016) and were 2010–2016 for trend analysis. The regression line in all periods showed a slight decreasing trend with slope of -0.036 ng m$^{-3}$ yr$^{-1}$ ($p = 0.0145$), whereas the regression line in 2010–2016 showed a slight increasing trend with a slope of 0.105 ng m$^{-3}$ yr$^{-1}$ ($p < 0.001$).

[Figure]

**Figure S4: Monthly long-term of EBC concentrations at Syowa Station, Antarctica in 2005–2016.**

[Figure]

**Figure S4: (continued)**

[Figure]

**Figure S5: Seasonal features of contributions of BC origins and their PSA. In the figure, AFN, AFS, AMM, AMN, AMS, AUS, EUR, CHN, IDN, IND, JPN, and SBR respectively denote northern Africa, southern Africa, Central America, North America, South America, Australia, Europe, China, Indonesia, India, Japan, and Siberia (as shown in Figure 2).**

[Figure]

**Figure S6: Seasonal variation of concentrations of oxalate and CH₃SO₃⁻ in aerosols at Syowa Station, Antarctica.**

Non-size-segregated aerosol sampling was done at Syowa Station, Antarctica. Aerosol sampling in 2003 and 2004–2006 was conducted, respectively, at an atmospheric observatory and clean air observatory. The clean air observatory was built in January 2004. Aerosol sampling was controlled using a wind selector to avoid local contamination. Sampling and analytical procedures were applied in accordance with Hara et al. (2004, 2010).

[Figure]

**Figure S7: Absorbance of aqueous solution (ca. 3 M) of CH₃SO₃H.**

Absorbance (optical absorption) of $CH_3SO_3H$ aqueous solution was found using a spectrophotometer with 5 nm bandwidth and light wavelength accuracy of ±2 nm (Genesis 30; Thermo Scientific). The measurable wavelength range is 325–1000 nm. Before determination, $CH_3SO_3H$ was diluted to ca. 3 M using ultrapure water.

---

## Author Comment (AC2) · 18 Mar 2019

Reply to comments from Referee #2: We would like to thank your helpful and complaisant comments to improve our manuscript. All comments are responded and addressed in the revised manuscript. Details are listed as follows. Corrected parts on comments from Ref. #2 were marked by blue characters in the revised manuscript.

Comment from Referee: I would suggest authors provide a clear explanation of data screening procedure (from measured data to useful data for BC).

Reply from Authors: Specific procedures for data screening were added into Section of 2 (Measurements, modelling and analysis).

Comment from Referee: I found abstract and conclusion a little confusing, vaguely

written, especially with the division of potential source areas (PSA), types of aerosols, and the CHASER model part is not mentioned explicitly.

Reply from Authors: Based on comment from you and referee #1, sentences in abstract were modified to simple and clear explanation.

Comment from Referee: While looking it at it, I got the first impression as high variability in daily EBC values between the years 2005 to 2009, which diminishes after that. It might be possible that more local influences during those years than in other years. I would suggest discussing if any change of location of instrument occurred. Is aethalometer calibrated? Or anything which authors wish to comment on that? Is the sampling line is heated or any changes?

Reply from Authors: At a glance, high EBC concentrations were often observed in winter–spring during 2005–2009. Measurement conditions (e.g., tube length and room temperature) and analytical procedures were the same from 2005–2016. Therefore, this change might result from variations of frequency or strength of EBC transport events rather than measurement and analytical reasons. These explanations were added to discussion in Fig. 3.

Comment from Referee: Air mass history and classification are explained very nicely, but it stands suddenly out of context. Authors have dedicated Figure 5, Figure 6 and Figure 7 especially for explaining this and they only connect it with EBC in Figure 8. I would suggest the authors make a good connection in meteorology and aerosol in these figures and sections. Also, classifications and their sub-classifications are not explained and connected for better readability.

Reply from Authors: Based on comment from you and referee #1, some statements were added in these explanations.

Comment from Referee: Based on the CHASER model, EBC origins were classified into three sections (biomass, fossil fuel, and other combustion). Later it is stated that

"other combustion" is broadly biomass burning, which makes it as two classifications, which could be inferred from angstrom absorption exponents (AAE) of Figure 3. Is it possible to add information about the mixed state (Internal or external) or aging (fresh or aged) of BC, used in the CHASER model, which showed promising results in monthly values and seasonality of BC in Figure 9?

Reply from Authors: CHASER model indicated that internal mixing state of EBC was dominant. The short explanation was added in the text of the revised manuscript. Details of model results will be published in another paper (now on preparation for publication).

Comment from Referee: At many places, authors replace "Syowa" with "coastal Antarctica" or "Antarctica coast". It might be useful to replace "Antarctica" in the title with "coastal Antarctica". But I would leave this totally on author's choice.

Reply from Authors: At some places, "Syowa" were changed "coastal Antarctica". In title, we changed to "coastal Antarctica".

Comment from Referee: Page 1: Line 10: We measured equivalent black carbon (BC) [ I think you measured BC and corrected it to make it EBC)

Reply from Authors: This was changed based on your suggestion.

Comment from Referee: Line 10: Feb 2005 to Feb 2016 [ adding the end month of measurement]

Reply from Authors: This was changed based on your suggestion.

Comment from Referee: Line 25: First statement needs a reference

Reply from Authors: A few references (e.g., Gelencsér, 2004; Gilardoni, and Fuzzi, 2017) were added in the text.

Comment from Referee: Line 34: Antarctic regions; BC concentrations

Reply from Authors: This was changed based on your suggestion.

Comment from Referee: Line 34: Page 2: The Antarctic is referred to here as one of the remote regions. I think calling it as "Antarctic region" throughout the manuscript is appropriate than "Antarctic regions"? I also think it is worth mentioning and dividing Antarctica as Eastern and Western Antarctica in the introduction as this section talks about tourism and transport from South America and the African region

Reply from Authors: "Antarctic regions" was changed to "Antarctic region" in the text. We added short statement using Eastern and Western Antarctica in the text.

Comment from Referee: Line 14: please specify where in Antarctica?

Reply from Authors: We added specific locations in the text.

Comment from Referee: Page 3: Section 2 heading could be "Measurements, Modelling, and Analysis"

Reply from Authors: This was changed based on your suggestion.

Comment from Referee: Line 5: "Research" is missing in JARE expansion

Reply from Authors: Exactly! This was changed based on your suggestion.

Comment from Referee: Line 6: It would be worthy to mention the altitude of sampling station at Syowa with latitude and longitude

Reply from Authors: In addition to latitude and longitude of Syowa Station, altitude (elevation) was added in the text.

Comment from Referee: Line 7 To Syowa, the icebreaker ship Shirase approaches every summer (mainly between end-December to – early February) for the transportation of fuel and materials for wintering operations and scientific activity

Reply from Authors: This was changed based on your suggestion.

Comment from Referee: Line 21: I think data screening procedure needs more clarity

as it is not in Hara et al., 2010. Hara et al., 2010 cites Hara et al., 2008 and I would suggest citing the right paper here.

Reply from Authors: As mentioned above, we added specific procedures for data screening in the text.

Comment from Referee: Line 23 to 34: what is the value of multiple scattering and loading parameter used in making BC to EBC? Would be a good idea to mention it explicitly here.

Reply from Authors: Specific parameters for BC correction by Weingartner's method were given in statements in Supplementary. To avoid confusion, we added statement of "correction factor for multiple scattering of light and shadowing effects in Weingartner's correction" in the Supplementary.

Comment from Referee: Line 28: Attenuation at 880 nm is used widely for BC re-trievals, I would suggest making changes in the statement accordingly

Reply from Authors: This was changed based on your suggestion.

Comment from Referee: Line 35: the statement "We use a multi-wavelength..." could be rephrased like "using the spectral (or multiwavelength) aerosol absorption values retrieved from aethalometer, we estimated AAE

Reply from Authors: This was changed based on your suggestion.

Comment from Referee: Page 4: Line 10-14: CHASER could be expanded. There are other acronyms also need to be expanded. I think it would be helpful for readers who are not modelers.

Reply from Authors: Acronyms in explanation on CHASER model were defined in the text and added into the Acronym list in Appendix.

Comment from Referee: Line 39: "cooking" not "cocking"

[Figure]

Reply from Authors: This was changed based on your suggestion.

Comment from Referee: Page 5: Section heading "Discussions"

Reply from Authors: Because "Discussion" is used usually in paper, we use "Discussion".

Comment from Referee: Line13-14: Any seasonal long-term trend at Syowa? It would be worth seeing whether there is an increase or decrease in summer (like Neumayer) or spring

Reply from Authors: Thank you very much for nice comment. We checked long-term trend in each month, as shown in Fig. S4. Trends were not clear expect July. The following explanation and discussion were added into the revised manuscript.

Although a decreasing trend of EBC concentrations in summer (November and December) was found at Neumayer (Weller et al., 2013), no seasonal long-term trend was clear at Syowa except for July (Fig. S4 in Supplementary Information). At a glance, EBC concentrations in July showed an increasing trend for 2011–2016 (0.325 ng m-3 yr-1 in monthly median and 0.363 ng m-3 yr-1 in monthly mean). However, we must consider the likelihood that EBC concentrations in winter (June–August) declined in 2010–2012 rather than following the increasing trend by EBC emissions at middle and low latitudes. Indeed, this variation in July might be related to changes of air mass origins (details are discussed in section 3.2).

Particularly, the contribution of transport from continental FT in March–October was higher than that in other years. This change corresponded to lower EBC concentrations in July of 2010–2012, as described above. Therefore, the increasing trend of EBC concentrations in July of 2010–2016 might not be a long-term trend but a temporal trend resulting from year-to-year variations of air mass history.

Comment from Referee: Line 25-42: It is not much clear. I would suggest defining seasons and maintain uniformity in the discussion of seasonality and comparison with

other stations. The possible sources in each season could be also be highlighted.

Reply from Authors: Specific months were added in the text, instead of definition of seasons.

Comment from Referee: Page 6: Line 21: BrC Brown Carbon (BrC)

Reply from Authors: BrC was defined in Introduction. Thus, we did not change it, here.

Comment from Referee: Line 29: larger negative values (<-0.4)

Reply from Authors: This was changed based on your suggestion.

Comment from Referee: Line 24: What is a high correlation means here? R2 values are lesser for June-Aug, in comparison to other months

Reply from Authors: We added explanation of "Particularly, high correlation (R2 > 0.7) was obtained in March, September-December."

Comment from Referee: Page 7: Line 6: Slopes >1 but AAE was lesser in spring (Figure 3d), so how you suggest it is biomass burning aerosol of organic origin? Please clarify

Reply from Authors: The following explanation was added to the revised manuscript.

The concentrations of EBC and organic aerosols derived from biomass burning increased in the spring maximum as described above, whereas the EBC concentrations decreased and the concentrations of organic aerosols such as CH3SO3- derived from oceanic bioactivity increased during summer.

Comment from Referee: Line 15: "First, we compare EBC data to the air mass history at Syowa" this line does not seem appropriate here

Reply from Authors: The statement of "First, we compare EBC data to the air mass history at Syowa" was removed from the text.

Comment from Referee: Line 17: for the 3rd classification, do you mean outflow from

the high-latitude Antarctic continent to coastal Antarctica? Please clarify

Reply from Authors: This explanation was changed to "(3) outflow from the high-latitudinal Antarctic continent to the coasts."

Comment from Referee: Line30: It appears that the probability density of air mass arriving at Syowa shows an East-West spread from one month to another month, as compared to North-South spread. In that case, transport from inland Antarctica is more important than long-range transport from populated continents. Is it the case?

Reply from Authors: This explanation was changed to "This difference implies that the transport strength of the outflow from the Antarctic continent had remarkable seasonal change in addition to important contribution of the poleward flow patterns from the ocean."

Comment from Referee: Line 41: could specify an approximate tropopause height.

Reply from Authors: We checked approximate tropopause height from previous work by Tomikawa et al. (2009). The sentence was modified in the revised manuscript, as follows.

Considering tropopause height (8–10 km) identified by O3 profiles in the Antarctica during the winter (Tomikawa et al., 2009), the air mass history implies that air masses near tropopause over the continent can flow to the boundary layer (BL) at the Antarctic coasts during winter.

Comment from Referee: Line 35-43: From Figure 6, it appears that Syowa is influenced by high-latitude inland Antarctica air mass during all months (which is relatively less in January). What is the final take from Figure 6?

Reply from Authors: Fig. 6 (Fig. 5 in the revised manuscript) indicates vertical motion of air mass in each latitude. Vertical mixing was varied largely depending on latitudes. This was already stated in the text. Some explanation was added in this section.

Comment from Referee: Page 8: Line 3-5: Is this subclassification of the classification on page 7(Line 16-18)? I think it is 2 sub-classifications of the previous classification, but it is not clear in the text.

Reply from Authors: To make clear explanation, the statement was changer as follows.

With suggestion of vertical motion and geographical classification of air mass origins as described above, the following transport patterns and air mass origins at Syowa are finally classifiable in this study: (1) poleward flow from MBL, (2) poleward flow from LFT, (3) westward flow along the coastal line via BL, (4) westward flow along the coastal line from LFT, (5) outflow from the FT over the Antarctic continent, and (6) outflow from BL over the Antarctic continent.

Comment from Referee: Line 6: classification of air mass origin > 75S could be re-named as remote continental or Antarctic continental, as naming it continental confuses with polluted and populated continents.

Reply from Authors: This was changed based on your suggestion (we used "Antarctic-continental"). Because of suggestion by referee #1, this classification was moved to Section 2.2.

Comment from Referee: Line 8: statement is not clear

Reply from Authors: From comments from you and referee #1, we add explanation to classify air mass origins. This explanation was also added in Section 2.2.

Comment from Referee: Line 29-30: Is an increase in MBL air mass origin EBC, could be due to Ship emissions in the Antarctic Circle (for fishing or tourism)?

Reply from Authors: Ship operation can emit BC (or EBC) to the atmosphere. However, the contribution may be negligible due to lower density of marine traffic in the Southern Ocean and near the Antarctic coasts. We add the following discussion and explanation in the revised manuscript.

Although high EBC concentrations were obtained in air masses from MBL, we must consider EBC origins in air masses from MBL. Additionally, 120-hr backward trajectory analysis was too short to reach to contributable PSA because it took longer than one week for transport from the coasts of South America and southern Africa to Syowa (Hara et al., 2010). Density of marine traffic (i.e. ship operation) in the Southern Ocean and near the Antarctic coasts was too low to engender an increase of EBC concentrations in air mass from MBL, although ship emissions can have an influence locally on EBC concentrations, for example ship-borne tourism in the Antarctic Peninsula during summer.

Comment from Referee: Page 9: Line 25-26: Filter biased problems and related uncertainty were not discussed while detailing EBC. I am glad that the authors bring it up here.

Reply from Authors: Discussion on filter biased problems and related uncertainty were added to sections of 2-1 (Page 3) and 3.1 (Page 6-7).

Comment from Referee: Line 28-29: This is already discussed in section 3.1

Reply from Authors: This sentence was removed in the revised manuscript.

Comment from Referee: Line 34-36: So basically, this classification is biomass and fossil fuel? As authors said other combustion is also biomass in the broad sense, so how this is different than AAE of section 3.1, besides it is from CHASER model?

Reply from Authors: We showed seasonal features of aerosol optical properties such as AAE. As shown in the manuscript, features of AAE was influenced by organic aerosols derived from combustion (dominantly biomass burning) and oceanic bioactivity, in addition to mixing states of EBC. Also, CHASER model provided us important knowledge on EBC origins and PSA. However, relation between AAE and each EBC origin contribution was not clear. This might result from the dominant contribution of biomass burning on EBC. Therefore, we did not add description about this in the revised manuscript.

Comment from Referee: Line 37-39: As the text says biomass burning is dominant in spring, and Figure 8 says it is Marine BL and Marine FT contributing to the EBC at Syowa, so what would be the conclusion? It is not clear

Reply from Authors: MBL and MFT were just transport pathway from EBC-PSA to Syowa. To avoid confusion and miss-understanding, discussion on origins of EBC in MBL was added in Section 3.3 in the revised manuscript.

Comment from Referee: Page 10: Line 1: It is difficult to identify the August-October peak in Figure 10b

Reply from Authors: To identify easily months, minor ticks every two months were added in Fig. (Fig. 9 in the revised manuscript).

Comment from Referee: Line 6: "contribution of BB" or "contribution from BB". Also, the difference is significant if you compare magnitudes of BC, which showed a 25 to 50 % decrease from 2011-2012 to 2015-2016

Reply from Authors: Magnitude of their contributions was added in the text, as follows.

The contributions of BB in South America and southern Africa in August–November were, respectively, 18.1–62.3% (mean 42.1%) and 15.9–71.7% (mean 43.3%). Relative importance of BB in South America and southern Africa showed a slight year-to-year difference.

Comment from Referee: Line 24-25: I am not clear about the statement which ends with a question mark. The explanation is given in the next lines and I consider that statement as a misfit. Is it possible to use a symbol for "BB-model-BC concentrations", something like the BB (BC) model? Similarly, for FFC and OC

Reply from Authors: Here, we changed the description to "We need to know transport pathway from Australia to the Syowa to understand the high BB-mBC concentrations

in Australia."

Comment from Referee: Line 30-35: It is quite difficult to follow the month to month explanation from Figure 10. Authors should either include minor labels or ticks or any other way to the identification The authors might consider a stacked column chart (by normalizing it with total concentrations) for all panel 10b, c, and d. So, the stacked column length would be total, partitions in the column would represent the contributions of South America, Southern Africa, and Australia. I think in that way, all the description in the text would be clearer. But I leave this to authors.

Reply from Authors: We need information about mBC concentrations to explain seasonal features. Seasonal features of contributions of BC origins and PSA were shown in Fig. S5. Therefore, we did not change these figures.

Comment from Referee: Page11: Line 1-6; I think as the southern America coastline extends much to the southern latitudes (near to western Antarctic peninsula), and the westward transport along coastal Antarctica, might be also a reason for the higher influence on the Antarctic BC, in addition to the GDP. This could be also clarified and detailed in the manuscript.

Reply from Authors: Based on your comment, the following description was added into discussion.

The relevant likelihoods must be discussed to elucidate this difference: (1) difference of transport pathway of anthropogenic EBC from South America and southern Africa to the Antarctica and (2) differences of EBC emission from anthropogenic combustion (i.e. fossil fuel use) in South America and southern Africa. Because of eastward cyclone movement in the Southern Ocean, air masses outflowed eastwardly from the continents of South America and southern Africa. Unlike the Africa continent, the South American continent extends to ca. 55ïĊřS. This geographical difference can engender higher contributions of anthropogenic EBC emitted from South America. Indeed, direct evidence of EBC transport from South America was reported in earlier works (Pereira

et al., 2006; Fiebig et al., 2009; Hara et al., 2010). In addition, higher contributions of South America were observed in transport of mineral dusts to the Antarctica (e.g., Delmonte et al., 2004, 2008; Gassó et al., 2010; Li et al., 2010).

Comment from Referee: Conclusion section: It should be rephrased to highlight important data set period, seasonality of BC, transport patterns at Syowa, model comparison and regional contribution from South America, South Africa, and Australia.

Reply from Authors: From suggestion from you and referee#1, statements in conclusion were modified to understand easily highlight results.

Comment from Referee: Figure 1: I think it would be better to place Syowa station as a different symbol or by placing the name next to the current symbol. Identifying regions like South America and Africa could be also a good idea as it comes quite often in the manuscript

Reply from Authors: Names of each station and others were labeled in Fig. 1.

Comment from Referee: Figure 2: I would suggest adding first and last labels in the Y axis too. Authors may consider writing Syowa near the red circle.

Reply from Authors: We added label in y-axis and name of Syowa in Fig. 2.

Comment from Referee: Figure 3: Y-axis scale for panel c is missing. It doesn't seem matching with panel a. Blue line mentioned in the caption is not visible in the panel a

Reply from Authors: Scales of y-axis were incorrect. These figures were modified. Also, blue line was added in the figure.

Comment from Referee: Figure 5: Syowa location could be shown in a different color/symbol for better visibility. I would suggest using a latitude scale too for this figure

Reply from Authors: Symbols for location of Syowa station were changed. Latitude scale was added in one of figure (not all fig.).

Comment from Referee: Figure 9: In panel b, the regression coefficient could be shown

Reply from Authors: Regression coefficient and relation (equation) were given in the text. Thus, we did not add them in Fig. 8 (in the revised manuscript).

Comment from Referee: Figure 10: Caption for the panels are not clear. Do authors mean BB aerosols from South America as a whole or BB from South America, contributing to EBC at Syowa.

Reply from Authors: The caption of Fig was modified in the revised manuscript as follows.

Figure 9: Seasonal features of (a) contribution of potential origins of mBC at Syowa Station, (b) the concentrations of mBC released from biomass burning in major PSA, (c) the concentrations of mBC released from combustion of fossil fuels in major PSA, and (d) the concentrations of mBC released from the others in major PSA.

Comment from Referee: List of Acronyms: Some Acronyms from the manuscript are missing in the list, like HYSPLIT, CHASER.

Reply from Authors: We added them and others into List of Acronyms.

Please also note the supplement to this comment:
https://www.atmos-chem-phys-discuss.net/acp-2018-1190/acp-2018-1190-AC2-supplement.pdf

[Figure]

**Supplement:**

[revised manuscript text omitted]

$$R = \frac{1}{\left(\frac{1}{f}-1\right)\frac{lnATN-\ln(10\%)}{ln(50\%)-ln(10\%)}+1} \tag{4}$$

Correction parameter $f$ is estimated using the following relation.

$$f = a(1-\omega_0)+1 \tag{5}$$

In that equation, $\omega_0$ is the single-scattering albedo. The determined parameters *a* are 0.87 ($\lambda = 450$ nm) and 0.85 ($\lambda = 660$ nm). Because of the low BC concentrations detected at Antarctic coasts, $\omega_0$ was found mostly as 0.97–0.99 at Syowa Station (Yabuki et al., preparation for publication). Similar values of $\omega_0$ were measured also at Neumayer (Weller et al., 2013). Therefore, *f* values can be 1.02–1 using our measurement conditions. Here, we used 1.01 as the *f* values. Furthermore, $\Delta t$ was 120 min (2 hr) in this study because of lower BC concentrations. The corrected BC concentrations were estimated for all BC data (recorded every 15 min) in this study. In aethalometer measurements using AE31, $\sigma_{ATN}$ is 14,625 nm m$^2$ g$^{-1}$ $\lambda^{-1}$.

[Figure]

**Figure S1: Histogram of correction factor of Weingartner's correction in aethalometer data measured at Syowa Station, Antarctica during our measurements.**

[Figure]

**Figure S2: Histogram of EBC concentration measured at Syowa Station, Antarctica in 2005–2016.**

[Figure]

**Figure S3: Variations of monthly median of EBC anomaly at Syowa Station, Antarctica. Red, blue, and green lines respectively present regression lines in all periods (2005–2016) and 2010–2016, and anomaly value of 1.**

In our EBC dataset, long gaps in data exist because of instrumental troubles in 2007 and Jan. – March in 2011; short data gaps were caused by local contamination. In addition, EBC concentrations showed strong seasonal variations as depicted in Fig. 3. First, one must remove the strong seasonality before analysis of long-term trends. Daily median EBC concentrations during our measurements were estimated as normal-like values of EBC concentrations at Syowa Station. Then, the ratios of ambient EBC concentrations to the normal-like EBC concentrations were calculated and were used as anomaly values. The monthly median EBC anomaly is shown in Fig. S3. Linear regression lines were fitted during all periods (2005–2016) and were 2010–2016 for trend analysis. The regression line in all periods showed a slight decreasing trend with slope of -0.036 ng m$^{-3}$ yr$^{-1}$ ($p = 0.0145$), whereas the regression line in 2010–2016 showed a slight increasing trend with a slope of 0.105 ng m$^{-3}$ yr$^{-1}$ ($p < 0.001$).

[Figure]

**Figure S4: Monthly long-term of EBC concentrations at Syowa Station, Antarctica in 2005–2016.**

[Figure]

**Figure S4: (continued)**

[Figure]

**Figure S5: Seasonal features of contributions of BC origins and their PSA. In the figure, AFN, AFS, AMM, AMN, AMS, AUS, EUR, CHN, IDN, IND, JPN, and SBR respectively denote northern Africa, southern Africa, Central America, North America, South America, Australia, Europe, China, Indonesia, India, Japan, and Siberia (as shown in Figure 2).**

[Figure]

**Figure S6: Seasonal variation of concentrations of oxalate and CH₃SO₃⁻ in aerosols at Syowa Station, Antarctica.**

Non-size-segregated aerosol sampling was done at Syowa Station, Antarctica. Aerosol sampling in 2003 and 2004–2006 was conducted, respectively, at an atmospheric observatory and clean air observatory. The clean air observatory was built in January 2004. Aerosol sampling was controlled using a wind selector to avoid local contamination. Sampling and analytical procedures were applied in accordance with Hara et al. (2004, 2010).

[Figure]

**Figure S7: Absorbance of aqueous solution (ca. 3 M) of CH₃SO₃H.**

Absorbance (optical absorption) of $CH_3SO_3H$ aqueous solution was found using a spectrophotometer with 5 nm bandwidth and light wavelength accuracy of ±2 nm (Genesis 30; Thermo Scientific). The measurable wavelength range is 325–1000 nm. Before determination, $CH_3SO_3H$ was diluted to ca. 3 M using ultrapure water.